# GIFT: UNLOCKING FULL POTENTIAL OF LABELS IN DISTILLED DATASET AT NEAR-ZERO COST

**Xinyi Shang**[1,*]    **Peng Sun**[2,3*]    **Tao Lin**[3,†]
[1]University College London    [2]Zhejiang University    [3]Westlake University
xinyi.shang.23@ucl.ac.uk, sunpeng@westlake.edu.cn, lintao@westlake.edu.cn

## ABSTRACT

Recent advancements in dataset distillation have demonstrated the significant benefits of employing soft labels generated by pre-trained teacher models. In this paper, we introduce a novel perspective by emphasizing the full utilization of labels. We first conduct a comprehensive comparison of various loss functions for soft label utilization in dataset distillation, revealing that the model trained on the synthetic dataset exhibits high sensitivity to the choice of loss function for soft label utilization. This finding highlights the necessity of a universal loss function for training models on synthetic datasets. Building on these insights, we introduce an extremely simple yet surprisingly effective plug-and-play approach, GIFT, which encompasses soft label refinement and a cosine similarity-based loss function to efficiently leverage full label information. Extensive experiments indicate that GIFT consistently enhances state-of-the-art dataset distillation methods across various dataset scales, without incurring additional computational costs. Importantly, GIFT significantly enhances cross-optimizer generalization, an area previously overlooked. For instance, on ImageNet-1K with IPC = 10, GIFT enhances the state-of-the-art method RDED by 30.8% in cross-optimizer generalization [1].

## 1 INTRODUCTION

Dataset distillation (DD) (Wang et al., 2018) has demonstrated its potential to significantly reduce data size while maintaining comparable model performance (Cazenavette et al., 2022; 2023; Zhao et al., 2020; Zhao & Bilen, 2023; Zhao et al., 2023). Most existing DD methods focus on optimizing images (Wang et al., 2018; Zhao & Bilen, 2022; Cazenavette et al., 2022; Kim et al., 2022; Liu et al., 2022), but recent studies (Yin et al., 2023; Sun et al., 2024; Shao et al., 2024; Guo et al., 2024) have highlighted the substantial benefits of soft labels. These studies utilize labels assigned by pre-trained models (also called teacher models), yielding siginificant enhancement in stability during the synthesizing process and considerable performance. Moreover, a notable study by Qin et al. (2024) examines the role of soft labels in depth, emphasizing their importance.

To fully explore the utilization of soft labels in the state-of-the-art dataset distillation methods, we conduct a comprehensive comparison of loss functions for soft label in synthesic datasets of IPC = 10 [2] via the SOTA dataset distillation methods on Tiny-ImageNet and large-scale ImageNet-1K [3]. As shown in Figure 1 , different dataset distillation methods employ distinct loss functions, and performance varies significantly across loss functions. In particular, simply replacing KL divergence loss (Hinton et al., 2015) with Soft cross-entropy loss (Bridle, 1989) for SRe$^2$L results in a significant performance drop of 13.3% on Tiny-ImageNet. This substantial gap indicates that models trained on synthetic datasets are ***highly sensitive*** to the choice of loss function.

Furthermore, we observe a notable performance degradation in these loss functions when applied ***across different optimizers***. For example, when utilizing distilled data produced by RDED and

---

[*]Equal contribution with the first author

[†]Corresponding author.

[1]Our code is available at https://github.com/LINs-lab/GIFT. We also have provided Pytorch implementation code in Appendix F .

[2]Detailed experiments are elaborated in Section 3 .

[3]The synthetic dataset of DATM for ImageNet-1k is not provided, so comparing with DATM is impossible.

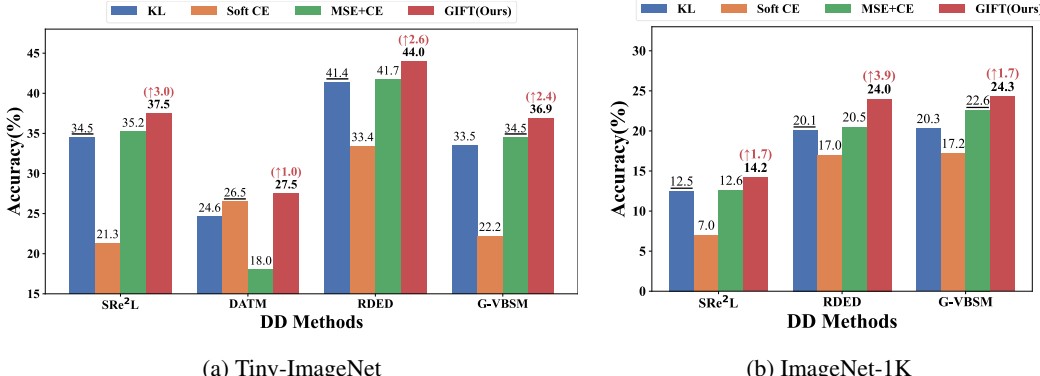

(a) Tiny-ImageNet          (b) ImageNet-1K

Figure 1: **Top-1 accuracy on various synthetic datasets via the SOTA dataset distillation methods across loss functions on Tiny-ImageNet and ImageNet-1K when `IPC =10`.** value means the results of the loss function used by the distillation method itself (e.g., SRe²L (Yin et al., 2023) uses KL divergence (Hinton et al., 2015)). **value** means the results of our GIFT, and (↑) denotes improvements over the dataset distillation methods. It is obvious that our method GIFT significantly enhances the dataset distillation methods.

training with the KL divergence loss, altering the optimizer from AdamW [4] to Adam results in a performance decrease from 47.5% to 17.8% (as shown in Table 8). *Hence, it is crucial to propose a universal and effective loss function that is robust across various scenarios.*

Moreover, one challenge of soft labels is that they are ***inherently suboptimal***, as the performance of the teacher model itself may be limited. For instance, a teacher model trained on ImageNet-1k on ConvNet has only 43.6% test accuracy. Despite its worse performance, such model is frequently used in Sun et al. (2024); Yin et al. (2023) to assign labels. Furthermore, in practical scenarios, it is common for the teacher model to have a smaller architecture than the student model due to cross-architecture challenge (Sun et al., 2024), which further limits the performance of the student model when only relying on an inferior teacher model. By contrast, hard labels are accurate and provide reliable supervision information. Nonetheless, directly utilizing hard labels through cross-entropy loss is not effective, as demonstrated in previous studies (Cui et al., 2023) and our experiments in Section 5.6. *Therefore, it is crucial to effectively integrate hard label information to mitigate the limitations associated with soft labels.*

Based on the above two findings, we propose an extremely simple yet effective plug-and-play approach called Gaining Improvement from Full Labels at Near-zero CosT (GIFT) to effectively utilize both hard and soft labels. It first refines soft labels by incorporating an additional smoothing label obtained through hard label smoothing (Szegedy et al., 2016). This simple module offers two significant advantages: firstly, it can correct erroneous signals from the teacher model, particularly in cases where the teacher assigns an incorrect label of the highest value; secondly, soft labels mainly contain intra-class information, which can hinder class separation to some extent (Zhang et al., 2015). Therefore, a relatively sharp label can enhance the dispersion between classes, thereby improving generalization ability, as demonstrated in Section 5.6. After obtaining the refined labels, we find that they are uniformly distributed and are approximately orthogonal to each other, as elaborated in Section 4. Therefore, we verify theoretically that simply using cosine similarity as the loss function achieves optimal performance. As shown in Figure 1 (red bars), our method GIFT consistently and significantly enhances the state-of-the-art dataset distillation methods across various scale datasets.

**In summary, our contributions are fourfold:**

(a) To the best of our knowledge, this paper is the first to provide a comprehensive comparison of loss functions for label utilization in dataset distillation. Our study reveals the intriguing fact that models trained on synthetic datasets are ***highly sensitive*** to the choice of loss function.

(b) We propose GIFT, a simple and universal label utilization algorithm including label refinement and a cosine similarity-based loss function. GIFT is built on top of the off-the-shelf dataset distillation methods and requires no extra information, thus raising ***no additional cost***. Moreover, we provide a theoretical analysis to support the proposed use of cosine similarity.

---

[4]RDED employs AdamW (Loshchilov, 2017) for evaluation.

(c) We identify a critical issue that has been overlooked in prior research: cross-optimizer generalization, as defined in Section 3.1 . We reveal that traditional loss functions suffer from significant robustness deficiencies when applied across different optimizers as detailed in Section 5.5 . In contrast, GIFT significantly enhances dataset distillation methods in cross-optimizer generalization. We conduct both empirical and theoretical analyses of this challenge in Appendix E .

(d) Experiments demonstrate that GIFT significantly improves performance over the state-of-the-art dataset distillation methods across varying scales and resolutions datasets, particularly for large-scale dataset distillation tasks. Furthermore, GIFT significantly enhances dataset distillation methods in cross-architecture, cross-optimizer generalization and proves advantageous in applications such as continual learning.

## 2 RELATED WORK

In dataset distillation, the majority of existing methods fix the labels as a one-hot format (Wang et al., 2018; Zhao & Bilen, 2022; Cazenavette et al., 2022; Liu et al., 2022; Zhao & Bilen, 2023), with a primary focus on optimizing synthetic images. Recent research highlights the significant benefits of utilizing soft labels to enhance model performance (Sun et al., 2024; Guo et al., 2024; Yin et al., 2023). Methods for obtaining soft labels can be broadly classified into two categories.

**Optimization-based Soft Labels.** The first type involves learning labels, with several studies (Bohdal et al., 2020; Nguyen et al., 2020; Sucholutsky & Schonlau, 2021; Zhou et al., 2022; Guo et al., 2024) finding that learning labels can significantly improve performance. Recent work (Guo et al., 2024) highlights that optimizing labels can enhance training stability and improve performance.

**Teacher model-based soft labels.** The subsequent works directly obtain soft labels. Inspired by knowledge distillation (Hinton et al., 2015), TESLA (Cui et al., 2023)introduces a soft label assignment strategy, directly generating soft labels by leveraging pre-trained teacher models trained on real datasets. These soft labels provide rich intra-class information, thereby improving distillation performance. Following this trend, the state-of-art methods (Yin et al., 2023; Sun et al., 2024; Shao et al., 2024) also utilize soft labels predicted by teacher models, achieving significant improvements.

## 3 MOTIVATION

### 3.1 PRELIMINARY

**Dataset Distillation.** Given a large dataset $\mathcal{D} = \{(\mathbf{x}_i, y_i)\}_{i=1}^N$, where $\mathbf{x}_i \in \mathbb{R}^d$ represents the input sample and $y_i \in \{1, \ldots, C\}$ denotes hard label, the objective of dataset distillation is to generate a synthetic dataset $\mathcal{S} = \{(\tilde{\mathbf{x}}_j, \tilde{y}_j)\}_{j=1}^M$, such that a model trained on $\mathcal{S}$ performs comparably to one trained on $\mathcal{D}$. We explore how to fully utilize both soft labels and hard labels in given datasets. Therefore, we re-define the synthetic dataset $\mathcal{S}$ as $\mathcal{S} = \{(\tilde{\mathbf{x}}_j, y_j, \tilde{y}_j)\}_{j=1}^M$, where $\tilde{\mathbf{x}}_j$ denotes the synthetic images, and $y_j$ and $\tilde{y}_j$ denote the corresponding hard labels and soft labels, respectively.

**Cross-Optimizer Generalization.** In deep learning, models use various network architectures and optimization algorithms. Optimizers like SGD and Adam have unique properties that affect model performance and generalization. Therefore, evaluating a distilled dataset's performance aross optimizers is essential to ensure its robustness across different training strategies.

> **Definition 1 (Cross-optimizer Generalization)** . *It refers to the capability of distilled datasets to maintain robust and consistent performance across different optimization algorithms.*

### 3.2 ARE LOSS FUNCTIONS PULLING THE STRINGS IN SYNTHETIC DATASET PERFORMANCE?

**Why do we need to answer this question?** Labels in dataset distillation are commonly and typically utilized through a variety of established loss functions, such as cross-entropy (CE) (Bridle, 1989), Kullback-Leibler (KL) divergence (Hinton et al., 2015), soft cross-entropy (Bridle, 1989) or mean squared error (MSE) (Nielsen, 2015). Specifically, the current state-of-the-art dataset distillation methods SRe$^2$L (Yin et al., 2023) and RDED (Sun et al., 2024) employ KL divergence, DATM (Guo et al., 2024) uses soft cross-entropy, and G-VBSM (Shao et al., 2024) simultaneously utilizes MSE and CE. However, different distillation methods employ varying loss functions to train models on synthetic datasets, yet there is a notable lack of comprehensive comparison among these loss

functions. Hence, we investigate the performance of four state-of-the-art dataset distillation methods under three commonly used loss functions.

**Experiment settings.** Our experiments span both small-scale Tiny-ImageNet and large-scale ImageNet-1K. Note that the synthetic dataset for DATM on ImageNet-1K is unavailable, preventing comparative analysis on this dataset. We conduct evaluations on these synthetic datasets with IPC $\in \{1, 10, 50\}$. Additional visualizations for Tiny-ImageNet and ImageNet-1K at IPC $= 1$ and IPC $= 50$ are provided in Appendix D . Notably, our empirical study is conducted from the end-user perspective: we treat the distillation of the synthetic dataset as a black box and apply different loss functions during the evaluation of the synthetic datasets.

**Results and Analysis.** The results, visualized in Figure 1 , reveal that even for the same synthetic dataset, the evaluation model performance varies significantly under different loss functions. For example, the performance of SRe$^2$L decreases by 13.2% on Tiny-ImageNet when switching from KL loss to soft CE loss but improves by 0.7% when using MSE+CE loss. These findings illustrate that the performance of models trained on synthetic datasets is ***highly sensitive*** to the choice of the loss function, *highlighting the necessity for a unified and effective loss function in dataset distillation.*

## 4 METHOD

Motivated by these findings, we propose an extremely simple but surprisingly effective plug-and-play approach called Gaining Improvement from Full Labels at Near-zero CosT (GIFT) to effectively utilize both hard and soft labels. GIFT includes two key modules: label refinement and a cosine similarity-based loss function. The former aims to refine soft labels by incorporating an additional smoothing label derived from hard label smoothing (Szegedy et al., 2016), while the latter is theoretically validated to achieve optimal performance simply using cosine similarity as the loss function. A PyTorch implementation of our method is provided in Appendix F .

**Label Refinement.** As discussed in Section 1 , soft labels generated by teacher models are inherently suboptimal due to two primary reasons. Firstly, the performance of the teacher model is limited, particularly on complex datasets such as ImageNet-1K. Secondly, soft labels predominantly provide intra-class information, thereby limiting class dispersion.

An intuitive method to address these shortcomings is to integrate with hard labels. On the one hand, hard labels offer accurate and reliable supervision, which can rectify erroneous information provided by soft labels. On the other hand, they can assist in inter-class dispersion. Therefore, we refine soft labels by weighing them with smoothed hard labels, thereby solving the two limitations of soft labels.

The refined soft label is defined as $\tilde{y}_j \leftarrow \gamma \cdot \frac{y_j}{\|y_j\|} + (1 - \gamma) \cdot \frac{\tilde{y}_j}{\|\tilde{y}_j\|}$, where $j$ is the j-th synthetic images, $y_j$ is the smoothed label obtained via label smoothing technique for hard label, and $\tilde{y}_j$ is the soft label. In our experiments, $\gamma = 0.1$ is validated to be optimal through experiments depicted in Figure 2 in Section 5.6 , and Figure 8 and Figure 9 in Appendix D .

**Mutual information bounded loss function.** Prior study (Sun et al., 2024) points that representation learning from any samples $X$ to targets $Y$ is based on maximizing their mutual information. They propose to distill the dataset by maximizing $I_\mathcal{V}(X, Y)$, where $I_\mathcal{V}$ denotes the $\mathcal{V}$-information (Xu et al., 2020), which has demonstrated superior performance. However, we observe that most prior studies, including (Sun et al., 2024), have not investigated training models using $I_\mathcal{V}(X, Y)$. To address this gap, we explore the application of $I_\mathcal{V}(X, Y)$ by deriving an upper bound for the $\mathcal{V}$-information, as presented in Theorem 1 (Detailed proof is provided in Appendix A ).

> **Theorem 1 .** *The $\mathcal{V}$-information $I_\mathcal{V}(X, Y)$ is upper bounded by a function involving the cosine similarity between the positive pair $(\mathbf{x}_i, y_i)$, the expected cosine similarity between the anchor $\mathbf{x}_i$ and negative samples $y_j$, and the number of negative samples $K$. Specifically,*
>
> $$\mathcal{L}_{\text{InfoNCE}} = -\mathbb{E}\left[\log \frac{\exp(f(\phi_\theta(\mathbf{x}), y))}{\sum_{y' \in \mathcal{Y}} \exp(f(\phi_\theta(\mathbf{x}), y'))}\right]$$
> $$\leq -\frac{1}{\tau}\left(\mathbb{E}\left[\left(\frac{\phi_\theta(\mathbf{x}_i) \cdot y_i}{\|\phi_\theta(\mathbf{x}_i)\|\|y_i\|}\right)\right] - \frac{\mathbb{E}[\phi_\theta(\mathbf{x}_i) \cdot y_j]}{\|\phi_\theta(\mathbf{x})_i\|\mathbb{E}[\|y_j\|]}\right) + \log(K),$$
>
> (1)

> *where $\tau$ denotes the temperature parameter, $\mathcal{L}_{\text{InfoNCE}}$ (Oord et al., 2018) serves as a proxy for $I_{\mathcal{V}}(X, Y)$ (Sun et al., 2024), and $f$ represents the similarity function.*

Moreover, the targets $Y$ in the synthetic dataset are pre-generated using pre-trained models (Sun et al., 2024; Yin et al., 2023). These targets can be considered high-dimensional vectors that are approximately orthogonal to each other (Ma et al., 2022; Yu et al., 2023; Awasthi et al., 2024). Consequently, the term $\mathbb{E}[\phi_{\boldsymbol{\theta}}(\mathbf{x}_i) \cdot y_j]$ approaches zero, allowing (1) to be simplified as:

$$\mathcal{L}_{\text{InfoNCE}} \leq -\frac{1}{\tau} \left( \mathbb{E}\left[ \frac{\phi_{\boldsymbol{\theta}}(\mathbf{x}_i) \cdot y_i}{\|\phi_{\boldsymbol{\theta}}(\mathbf{x}_i)\|\|y_i\|} \right] - \epsilon \right) + \log(N), \tag{2}$$

where $\epsilon$ is a small positive term approaching zero.

To minimize this upper bound (2), our proposed loss function for training on distilled data $\mathcal{S}$ can defined as:

$$\mathcal{L} = \mathbb{E}_{(\tilde{\mathbf{x}}_i, \tilde{y}_i) \sim \mathcal{S}} \left[ 1 - \frac{\phi_{\boldsymbol{\theta}}(\tilde{\mathbf{x}}_i) \cdot \tilde{y}_i}{\|\phi_{\boldsymbol{\theta}}(\tilde{\mathbf{x}}_i)\|\|\tilde{y}_i\|} \right], \tag{3}$$

where $\tilde{y}_j$ is the refined soft label discussed above and we replace the original training loss with (3).

## 5 EXPERIMENTS

In this section, we evaluate the superiority of our proposed GIFT across various datasets and architectures. First, we demonstrate the superior improvements of GIFT over the state-of-the-art dataset distillation in Section 5.2 and knowledge distillation methods in Section 5.3. Subsequently, we validate that GIFT achieves near-zero computational cost relative to baseline approaches (Section 5.4). Additionally, we show that GIFT significantly improves cross-architecture and cross-optimizer generalization (Section 5.5). To further understand the contributions of individual components, we perform detailed ablation studies (Section 5.6). Finally, we demonstrate the effectiveness of GIFT in enhancing continual learning performance, as detailed in Appendix D. For additional experimental details and comprehensive results, refer to Appendix C and Appendix D.

### 5.1 EXPERIMENT SETUP

**Datasets and Networks.** We conduct experiments on both large-scale and small-scale datasets, including the full $224 \times 224$ ImageNet-1k (Deng et al., 2009), Tiny-ImageNet (Le & Yang, 2015) and CIFAR-100 (Krizhevsky et al., 2009). Following previous dataset distillation studies (Yin et al., 2023; Cazenavette et al., 2022; Zhao et al., 2023; Cui et al., 2023; Guo et al., 2024), we employ ConvNet (Guo et al., 2024) and ResNet-18 (He et al., 2016) as our backbone architectures across all datasets. Specifically, Conv-3 is employed for CIFAR-10/100, while Conv-4 is used for Tiny-ImageNet and ImageNet-1K. For cross-architecture experiments, we additionally utilize large-scale networks, including ResNet-101 (He et al., 2016) and Swin-V2-Tiny (Liu et al., 2021) and small-scale networks, such as EfficientNet-B0 (Tan & Le, 2019), and MobileNet-V2 (Sandler et al., 2018) to verify the generalizability of our approach.

**Baselines.** We benchmark our method GIFT against state-of-the-art dataset distillation methods. We categorize current state-of-the-art methods based on two key factors: scalability to ImageNet-1K and the utilization of soft labels. These categorizations are summarized in Table 13 in Appendix C. Given that our primary focus is on enhancing the use of soft labels in dataset distillation, we restrict our comparisons to methods that leverage soft labels, including SRe$^2$L (Yin et al., 2023), RDED (Sun et al., 2024), DATM (Guo et al., 2024), G-VBSM (Shao et al., 2024), and CDA (Yin & Shen, 2023) Additionally, DATM only provides synthetic datasets on ConvNet and does not provide the ImageNet-1k synthetic dataset. CDA provides higher `IPC` synthetic datasets of Tiny-ImageNet and ImageNet-1k, distilled using ResNet architectures, so we mainly compare with CDA in Section 5.2.

Knowledge distillation (Hinton et al., 2015) is a straightforward approach that utilizes both hard and soft label information. Thus, we also compare our method GIFT with state-of-the-art knowledge distillation methods, including KD (Hinton et al., 2015), WSLD (Zhou et al., 2021), DKD (Zhao et al., 2022), and NKD (Yang et al., 2023). Further details on these methods are provided in Appendix C.

Table 1: **Comparison with the state-of-the-art methods of dataset distillation on CIFAR-100 and Tiny-ImageNet.** In this table, "-" are absent due to scalability.

| Network | Method | CIFAR100 | | | Tiny-ImageNet | | |
|---|---|---|---|---|---|---|---|
| | | 1 | 10 | 50 | 1 | 10 | 50 |
| ConvNet | SRe$^2$L | 13.6 ± 0.4 | 33.7 ± 0.5 | 52.3 ± 0.2 | 12.1 ± 0.4 | 34.5 ± 0.4 | 46.3 ± 0.1 |
| | SRe$^2$L + Ours | 15.1 ± 0.3 (↑ 1.5) | 38.0 ± 0.5 (↑ 4.3) | 55.4 ± 0.1 (↑ 3.1) | 13.1 ± 0.2 (↑ 1.0) | 37.5 ± 0.3 (↑ 3.0) | 47.1 ± 0.1 (↑ 0.8) |
| | RDED | 22.1 ± 0.3 | 47.5 ± 0.3 | 55.7 ± 0.4 | 17.9 ± 0.3 | 41.4 ± 0.3 | 47.2 ± 0.1 |
| | RDED + Ours | 24.7 ± 0.3 (↑ 2.5) | 50.6 ± 0.3 (↑ 2.5) | 57.9 ± 0.2 (↑ 2.2) | 19.1 ± 0.3 (↑ 1.2) | 44.0 ± 0.2 (↑ 2.6) | 48.3 ± 0.1 (↑ 1.1) |
| | DATM | - | 36.1 ± 0.2 | 43.0 ± 0.2 | - | 26.5 ± 0.2 | 34.2 ± 0.5 |
| | DATM + Ours | - | 37.8 ± 0.3 (↑ 1.7) | 43.6 ± 0.3 (↑ 0.6) | - | 27.5 ± 0.2 (↑ 1.0) | 34.8 ± 0.6 (↑ 0.6) |
| | G-VBSM | 14.7 ± 0.5 | 40.9 ± 0.4 | 54.7 ± 0.3 | 8.4 ± 0.4 | 34.5 ± 0.5 | 47.0 ± 0.3 |
| | G-VBSM + Ours | 16.0 ± 0.2 (↑ 1.3) | 44.6 ± 0.2 (↑ 3.7) | 57.2 ± 0.1 (↑ 2.5) | 8.9 ± 0.3 (↑ 0.5) | 36.9 ± 0.7 (↑ 2.4) | 47.8 ± 0.2 (↑ 0.8) |
| ResNet-18 | SRe$^2$L | 11.5 ± 0.5 | 42.7 ± 0.2 | 57.8 ± 0.6 | 12.7 ± 0.3 | 43.5 ± 0.1 | 53.9 ± 0.0 |
| | SRe$^2$L + Ours | 12.7 ± 0.4 (↑ 1.2) | 44.3 ± 0.3 (↑ 1.6) | 58.6 ± 0.3 (↑ 0.8) | 14.2 ± 0.3 (↑ 1.5) | 44.2 ± 0.3 (↑ 0.7) | 54.5 ± 0.2 (↑ 0.6) |
| | RDED | 4.7 ± 0.1 | 52.8 ± 0.2 | 64.4 ± 0.1 | 15.1 ± 0.3 | 48.2 ± 0.4 | 57.6 ± 0.3 |
| | RDED + Ours | 5.0 ± 0.2 (↑ 0.3) | 54.0 ± 0.3 (↑ 1.2) | 65.3 ± 0.2 (↑ 0.7) | 15.9 ± 0.3 (↑ 0.8) | 49.2 ± 0.1 (↑ 1.0) | 58.1 ± 0.1 (↑ 0.5) |
| | DATM | - | 25.8 ± 1.0 | 47.5 ± 0.4 | - | 26.7 ± 0.2 | 41.9 ± 0.3 |
| | DATM + Ours | - | 26.3 ± 0.4 (↑ 0.5) | 47.9 ± 0.3 (↑ 0.4) | - | 29.0 ± 0.5 (↑ 2.3) | 42.4 ± 0.2 (↑ 0.4) |
| | G-VBSM | 13.4 ± 0.3 | 48.5 ± 0.5 | 62.0 ± 0.2 | 8.8 ± 0.3 | 39.9 ± 0.4 | 52.8 ± 0.2 |
| | G-VBSM + Ours | 13.7 ± 0.3 (↑ 0.3) | 49.2 ± 0.2 (↑ 0.7) | 62.5 ± 0.3 (↑ 0.5) | 9.3 ± 0.3 (↑ 0.5) | 40.5 ± 0.2 (↑ 0.6) | 53.1 ± 0.1 (↑ 0.3) |

Table 2: **Comparison with the state-of-the-art methods of dataset distillation on ImageNet-1K.** In the table, (↑) means the improvements over these methods.

| | ImageNet-1K | | | | | |
|---|---|---|---|---|---|---|
| | ConvNet | | | ResNet-18 | | |
| Method | 10 | 50 | 100 | 10 | 50 | 100 |
| SRe$^2$L | 12.5 ± 0.3 | 35.4 ± 1.0 | 40.1 ± 0.4 | 31.5 ± 0.3 | 49.5 ± 0.1 | 54.3 ± 0.2 |
| SRe$^2$L + Ours | 14.2 ± 0.6 (↑ 1.7) | 38.1 ± 0.4 (↑ 2.7) | 41.5 ± 0.2 (↑ 1.4) | 31.9 ± 0.2 (↑ 0.4) | 50.1 ± 0.2 (↑ 0.6) | 54.8 ± 0.1 (↑ 0.5) |
| RDED | 20.1 ± 0.4 | 38.5 ± 0.2 | 41.8 ± 0.2 | 41.4 ± 0.4 | 55.5 ± 0.2 | 58.8 ± 0.1 |
| RDED + Ours | 24.0 ± 0.8 (↑ 3.9) | 39.5 ± 0.1 (↑ 1.0) | 42.5 ± 0.1 (↑ 0.7) | 43.2 ± 0.1 (↑ 1.8) | 56.5 ± 0.1 (↑ 1.0) | 59.3 ± 0.1 (↑ 0.5) |
| G-VBSM | 22.6 ± 0.5 | 37.3 ± 0.3 | 40.1 ± 0.4 | 36.7 ± 0.2 | 52.3 ± 0.1 | 57.3 ± 0.1 |
| G-VBSM + Ours | 24.3 ± 0.2 (↑ 1.7) | 39.1 ± 0.3 (↑ 1.8) | 42.1 ± 0.3 (↑ 2.0) | 37.9 ± 0.5 (↑ 1.2) | 53.1 ± 0.2 (↑ 0.8) | 57.6 ± 0.1 (↑ 0.3) |

**Implementation details of GIFT.** Our method does not involve any distilling datasets process. We obtain all synthetic datasets directly from the source data provided by the authors [5]. Notably, distilled data is generalized using both ConvNet and ResNet-18. We replace the loss function during evaluation. Thus, our method is a plug-and-play approach that can be easily integrated into existing dataset distillation pipelines without additional dataset synthesis or modification.

For the data augmentation of synthetic datasets, only synthetic datasets generated via DATM (Guo et al., 2024) are processed using ZCA whitening, as these datasets were initially distilled through ZCA whitening. Other distilled datasets are processed using DSA (Zhao & Bilen, 2021), as detailed in Table 14 . All experiments are conducted using an NVIDIA RTX 4090 GPU.

**Hyperparameter Settings.** We provide detailed hyperparameter configurations for our synthetic dataset evaluation in Appendix C . Following recent works (Yin et al., 2023; Shao et al., 2024), the evaluation on all datasets uses the parameters outlined in Table 15 . We set the coefficient of label smoothing $\alpha = 0.1$ and the weight hyper-parameter $\gamma = 0.1$ for all methods across various synthetic datasets, as $\gamma = 0.1$ is validated to be optimal through experiments depicted in Section 5.6 and Section 8 in Appendix D .

## 5.2 CAN GIFT IMPROVE PERFORMANCE OF DATASET DISTILLATION?

**Small-Scale Dataset Comparison.** In Table 1 , we present the test accuracy on CIFAR-100 and Tiny-ImageNet datasets before and after applying our GIFT algorithm. Notably, DATM does not provide synthetic datasets when IPC =1, nor does it provide synthetic datasets for ResNet-18. Consequently, we used ConvNet synthetic data to train ResNet-18. It is evident that *applying* GIFT *increases performance for all baseline methods*. Specifically, the SRe$^2$L method exhibits the most significant improvement, with an accuracy gain of up to 4.3% on CIFAR-100 when IPC =10. This is particularly noteworthy as GIFT requires no additional information or cost. The considerable accuracy gains can be achieved simply by replacing the loss function with our proposed approach.

---

[5] ⋆ SRe$^2$L: https://github.com/VILA-Lab/SRe2L
⋆ RDED: https://github.com/LINs-lab/RDED
⋆ DATM: https://gzyaftermath.github.io/DATM/
⋆ G-VBSM: https://github.com/shaoshitong/G_VBSM_Dataset_Condensation
⋆ CDA: https://github.com/VILA-Lab/SRe2L/tree/main/CDA

Table 3: **Comparison with CDA under higher `IPC` on small-scale Tiny-ImageNet and large-scale ImageNet-1K on ResNet-18.** In the table, (↑) means the improvements over CDA.

| Method | Tiny-ImageNet | | ImageNet-1K | | |
|---|---|---|---|---|---|
| | 50 | 100 | 50 | 100 | 200 |
| CDA | 49.5 ± 0.4 | 53.5 ± 0.3 | 53.7 ± 0.3 | 58.3 ± 0.3 | 63.4 ± 0.2 |
| CDA + Ours | 54.5 ± 0.3 (↑ 5.0) | 56.6 ± 0.2 (↑ 3.1) | 54.8 ± 0.2 (↑ 1.1) | 59.0 ± 0.2 (↑ 0.8) | 63.9 ± 0.1 (↑ 0.5) |

Table 4: **Comparison with RDED Large-scale Netwrok.** In the table, (↑) means the improvements over RDED.

| Method | Tiny-ImageNet | | ImageNet-1K | |
|---|---|---|---|---|
| | 10 | 50 | 10 | 50 |
| RDED | 47.1 ± 0.3 | 55.1 ± 0.3 | 42.3 ± 0.2 | 58.6 ± 0.1 |
| RDED + Ours | 48.6 ± 0.2 (↑ 1.5) | 56.2 ± 0.2 (↑ 1.1) | 43.5 ± 0.2 (↑ 1.2) | 59.4 ± 0.2 (↑ 0.8) |

Table 5: **Comparison with the knowledge distillation methods on the synthetic dataset via RDED (Sun et al., 2024) using ConvNet.** In this table, **bold** means the best result, underlined means the second best, and (↑) denotes improvements over the second best baseline.

| | CIFAR100 | | Tiny-ImageNet | | ImageNet-1K | | |
|---|---|---|---|---|---|---|---|
| | 10 | 50 | 10 | 50 | 10 | 50 | 100 |
| Teacher | 61.27 | 61.27 | 49.73 | 49.73 | 43.6 | 43.6 | 43.6 |
| KD | 43.2 ± 0.1 | 51.9 ± 0.4 | 33.3 ± 0.4 | 40.7 ± 0.2 | 16.7 ± 0.1 | 24.3 ± 0.2 | 27.5 ± 0.4 |
| WSLD | 38.6 ± 0.3 | 49.5 ± 0.5 | 27.1 ± 0.3 | 37.5 ± 0.2 | 13.4 ± 0.1 | 22.8 ± 0.1 | 26.2 ± 0.0 |
| DKD | 49.0 ± 0.2 | 56.7 ± 0.2 | 40.9 ± 0.2 | 47.2 ± 0.1 | 20.5 ± 0.1 | 33.1 ± 0.1 | 36.5 ± 0.1 |
| NKD | 46.3 ± 0.3 | 54.3 ± 0.1 | 37.6 ± 0.4 | 44.1 ± 0.2 | 19.84 ± 0.1 | 27.9 ± 0.2 | 30.6 ± 0.2 |
| GIFT (ours) | **50.6 ± 0.3** (↑ 1.6) | **57.9 ± 0.2** (↑ 1.2) | **44.0 ± 0.2** (↑ 3.1) | **48.3 ± 0.1** (↑ 1.1) | **24.0 ± 0.8** (↑ 3.4) | **39.5 ± 0.1** (↑ 6.4) | **42.5 ± 0.1** (↑ 6.0) |

**Large-Scale Dataset Comparison.** In the large-scale ImageNet-1k dataset, as reported in Table 2, our proposed method GIFT consistently improves all baseline methods across all `IPC` values in 10, 50, 100, using both ConvNet and ResNet-18 as evaluation models. Specifically, compared with the current state-of-the-art methods RDED and G-VBSM, GIFT achieves significant performance gains of 1.8% and 1.2% on ResNet-18 when `IPC` =10, respectively. The substantial performance improvements obtained by GIFT *demonstrate its capability to effectively scale to large-scale datasets.*

**Comparison under Higher `IPC`.** Given that only CDA (Yin & Shen, 2023) provides a higher `IPC` synthetic dataset distilled, our comparison primarily centers on CDA. The results in Table 3 demonstrate that our GIFT method substantially improves the performance of CDA. Furthermore, it also achieves significant enhancements at higher `IPC`.

**Comparison on Large-scale Network.** In addition to conventional networks like ResNet-18, we employ the state-of-the-art dataset distillation method, RDED, to generate distilled data for Tiny-ImageNet and ImageNet-1K using Swin Transformer (Liu et al., 2021). The results, as shown in Table 4, indicate notable enhancements, verifying the effectiveness and promise of our method.

## 5.3 CAN KNOWLEDGE DISTILLATION WORK?

A straightforward approach to combining soft labels and hard labels is knowledge distillation (Hinton et al., 2015), which transfers knowledge from a teacher model to a student model using hard labels (cross-entropy loss) and soft labels provided by a strong teacher model (KL divergence loss). In Table 5, we compare our proposed method, GIFT, with the state-of-the-art knowledge distillation techniques across synthetic datasets distilled via RDED (Sun et al., 2024) [6].

It can be observed that GIFT *outperforms all knowledge distillation methods*. We attribute the failure of knowledge distillation methods to the extremely small size of synthetic datasets, which significantly hampers the performance of knowledge distillation methods, as corroborated by (Stanton et al., 2021). Therefore, knowledge distillation is not well-suited for our problem, further highlighting the effectiveness of GIFT.

---

[6]RDED is the current state-of-the-art method as shown in Table 1, so we conduct the experiment on its synthetic datasets.

Table 6: **Training Time (s) and memory (GB) costs on three synthetic datasets using ResNet-18 when IPC=10.** (+) denotes additional cost over these methods.

| Method | CIFAR-100 | | Tiny-ImageNet | | ImageNet-1K | |
|---|---|---|---|---|---|---|
| | Training Time | Memory | Training Time | Memory | Training Time | Memory |
| SRe$^2$L | 180.22 | 0.69 | 1249.66 | 2.01 | 2011.24 | 2.32 |
| SRe$^2$L + Ours | 181.36 (+ 1.14) | 0.69 | 1275.75 (+ 26.09) | 2.01 | 2078.59 (+ 67.35) | 2.32 |
| RDED | 171.97 | 0.69 | 1272.28 | 2.01 | 2066.35 | 2.32 |
| RDED + Ours | 175.17 (+ 3.20) | 0.69 | 1298.73 (+ 26.45) | 2.01 | 2124.61 (+ 58.26) | 2.32 |
| DATM | 132.51 | 1.36 | 1018.31 | 12.55 | - | - |
| DATM + Ours | 139.06 (+ 6.55) | 1.36 | 1039.65 (+ 21.34) | 12.55 | - | - |
| G-VBSM | 183.90 | 0.69 | 1256.35 | 2.01 | 2074.72 | 2.32 |
| G-VBSM + Ours | 187.83 (+ 3.93) | 0.69 | 1282.63 (+ 26.28) | 2.01 | 2129.98 (+ 55.26) | 2.32 |

Table 7: **Top-1 accuracy on cross-architecture generalization on Tiny-ImageNet.** We use the synthetic datasets distilled (D) on ConvNet and ResNet-18 on Tiny-ImageNet when IPC=10. Then, evaluations (E) are performed across both small-scale and large-scale architectures. (↑) denotes improvements over these methods.

| | D/E | Small-Scale Architecture | | | | Large-Scale Architecture | |
|---|---|---|---|---|---|---|---|
| | | ConvNet | ResNet-18 | EfficientNet-B0 | MobileNet-V2 | ResNet-101 | Swin-V2-Tiny |
| ConvNet | SRe$^2$L | 34.5 ± 0.4 | 43.04 ± 0.1 | 11.2 ± 1.1 | 14.0 ± 0.6 | 9.9 ± 0.5 | 10.3 ± 0.4 |
| | SRe$^2$L + Ours | 37.5±0.3 (↑ 3.0) | 44.2 ± 0.3 (↑ 1.16) | 15.3 ± 1.0 (↑ 4.1) | 14.4 ± 0.3 (↑ 0.4) | 10.6 ± 0.3 (↑ 0.7) | 11.2 ± 0.2 (↑ 0.9) |
| | RDED | 41.4 ± 0.3 | 46.5 ± 0.2 | 30.3 ± 1.4 | 30.2 ± 0.2 | 28.2 ± 1.9 | 26.8 ± 0.6 |
| | RDED + Ours | 44.0±0.2 (↑ 2.6) | 47.2 ± 0.1 (↑ 0.7) | 31.4 ± 0.9 (↑ 1.1) | 31.3 ± 0.2 (↑ 1.1) | 30.8 ± 1.8 (↑ 2.6) | 28.7 ± 0.4 (↑ 1.9) |
| | DATM | 36.1 ± 0.2 | 27.3 ± 0.2 | 18.0 ± 0.3 | 14.7 ± 0.2 | 15.3 ± 0.8 | 4.1 ± 3.1 |
| | DATM + Ours | 37.8 ± 0.3 (↑ 1.7) | 29.0 ± 0.2 (↑ 1.7) | 18.4± 0.5 (↑ 0.4) | 17.5 ± 0.1 (↑ 2.8) | 16.8± 0.5 (↑ 1.5) | 16.3 ± 0.3 (↑ 12.2) |
| | G-VBSM | 34.5 ± 0.5 | 42.2 ± 0.3 | 13.7 ± 1.4 | 15.0 ± 0.4 | 7.7 ± 0.6 | 10.4 ± 0.4 |
| | G-VBSM + Ours | 36.9±0.7 (↑ 2.4) | 42.8 ± 0.2 (↑ 0.6) | 16.3 ± 1.0 (↑ 2.6) | 15.8± 0.4 (↑ 0.8) | 15.5 ± 0.3 (↑ 7.8) | 13.2± 0.1 (↑ 2.8) |
| ResNet-18 | SRe$^2$L | 19.2 ± 0.1 | 43.5 ± 0.1 | 11.6 ± 0.4 | 11.9 ± 0.3 | 8.7 ± 1.0 | 8.0 ± 0.2 |
| | SRe$^2$L + Ours | 19.4 ± 0.2 (↑ 0.2) | 44.2±0.3(↑ 0.7) | 12.2 ± 0.2 (↑ 0.6) | 12.3 ± 0.1 (↑ 0.4) | 9.0 ± 0.5 (↑ 0.3) | 8.8± 0.3 (↑ 0.8) |
| | RDED | 29.2 ± 0.3 | 48.2 ± 0.4 | 24.1 ± 0.7 | 23.5 ± 0.3 | 21.8 ± 0.3 | 19.6 ± 0.4 |
| | RDED + Ours | 29.9 ± 0.1 (↑ 0.7) | 49.2±0.1 (↑ 1.0) | 25.2 ± 0.2 (↑ 1.1) | 24.1 ± 0.3 (↑ 0.6) | 23.5± 0.3 (↑ 1.7) | 20.4± 0.3 (↑ 0.8) |
| | G-VBSM | 16.0 ± 0.3 | 39.9 ± 0.4 | 8.8 ± 0.1 | 11.5 ± 0.4 | 6.3 ± 1.1 | 6.5 ± 0.3 |
| | G-VBSM + Ours | 16.5 ± 0.4 (↑ 0.5) | 40.5±0.2 (↑ 0.6) | 10.1± 0.2 (↑ 1.3) | 11.8 ± 0.3 (↑ 0.3) | 9.2 ± 0.6 (↑ 2.9) | 8.2± 0.3 (↑ 1.7) |

## 5.4 CAN GIFT ACHIEVE NEAR-ZERO COST?

We perform experiments to evaluate memory and training time costs using ResNet-18 across datasets with varying scales and resolutions, as presented in Table 6. *Our method demonstrates no additional peak memory usage and incurs negligible computational overhead.* This efficiency is due to its emphasis on label refinement and the implementation of a general and simple loss function during the evaluation phases. Importantly, despite the negligible additional cost, GIFT yields significant performance improvements across datasets of varying scales and resolutions.

## 5.5 CAN GIFT IMPROVE GENERALIZATION?

**Cross-Architecture Generalization.** To validate the enhancement of generalization capability by our GIFT, it is necessary to assess its effectiveness across various neural architectures not encountered during the dataset synthesis phase. We evaluate performance on both small and large-scale model architectures. Table 7 presents the performance before and after applying our GIFT to dataset distillation methods. The results indicate that GIFT *enhances the cross-architecture generalization of all dataset distillation methods across diverse architectures*. Notably, our method shows significant improvements when generalizing from small networks to larger networks. For instance, GIFT yields performance gains of 2.6% and 7.8% for RDED and G-VBSM, respectively, when synthesizing data using ConvNet while training model on ResNet-101.

The success of our method is attributed to its stability. In cross-architecture scenarios, soft labels may not be sufficient due to architectural differences (Vyas et al., 2020). However, our cosine similarity approach inherently includes a normalization operation, mitigating the negative impact of label fluctuation. These results are promising, indicating that our method, which does not incur additional computational costs, is well-suited for applications involving large-scale models.

**Cross-optimizer Generalization.** Similar to cross-architecture generation, it is crucial to estimate the cross-optimizer generalization of dataset distillation methods. Different optimizers, such as

Table 8: **Top-1 accuracy (%) on cross-optimization generalization on Tiny-ImageNet and CIFAR100 when IPC =10.** We evaluate the performance of synthetic datasets across various optimizers.

| Dataset | CIFAR100 | | | Tiny-ImageNet | | |
|---|---|---|---|---|---|---|
| Optimizer | SGD | Adam | AdamW | SGD | Adam | AdamW |
| SRe$^2$L | 1.5 ± 0.1 | 8.7 ± 0.1 | 34.5 ± 0.4 | 0.6 ± 0.0 | 3.1 ± 0.1 | 33.7 ± 0.5 |
| SRe$^2$L + Ours | 43.0 ± 0.8 (↑ 42.6) | 44.7 ± 0.6 (↑ 37.5) | 38.0 ± 0.5 (↑ 3.5) | 43.8 ± 0.5 (↑ 43.2) | 45.2 ± 0.2 (↑ 42.1) | 37.5± 0.3 (↑ 3.0) |
| RDED | 1.9 ± 0.0 | 17.8 ± 0.1 | 47.5 ± 0.3 | 0.6 ± 0.0 | 4.5 ± 0.2 | 41.4 ± 0.3 |
| RDED + Ours | 53.4 ± 0.2 (↑ 51.5) | 53.7 ± 0.4 (↑ 35.9) | 50.6 ± 0.3 (↑ 2.6) | 46.6 ± 0.3 (↑ 46.0) | 46.5 ± 0.2 (↑ 42.0) | 44.0 ± 0.2 (↑ 2.6) |
| DATM | 37.3 ± 0.3 | 36.7 ± 0.1 | 36.1 ± 0.2 | 28.2 ± 0.1 | 27.8 ± 0.1 | 26.5 ± 0.2 |
| DATM + Ours | 38.5± 0.2 (↑ 1.2) | 40.0± 0.2 (↑ 3.3) | 37.8 ± 0.3 (↑ 1.7) | 30.1 ± 0.3 (↑ 1.9) | 29.1 ± 0.1 (↑ 1.3) | 27.5 ± 0.2 (↑ 1.0) |
| G-VBSM | 44.2 ± 1.8 | 41.2 ± 0.4 | 40.9 ± 0.4 | 41.4 ± 0.4 | 36.5 ± 0.4 | 34.5 ± 0.5 |
| G-VBSM + Ours | 49.8 ± 0.4 (↑ 5.6) | 51.3 ± 0.6 (↑ 10.1) | 44.6 ± 0.2 (↑ 3.7) | 44.5 ± 0.1 (↑ 3.1) | 45.3 ± 0.1 (↑ 8.8) | 36.9 ± 0.7 (↑ 2.4) |

Table 9: **Comparsion with loss functions employed in dataset distillation**. The experiment is conducted on synthetic datasets distilled via RDED (Sun et al., 2024). In this table, **bold** means the best result, underlined means the second best, and (↑) denotes improvements over the second best baseline.

| | | CIFAR100 | | Tiny-ImageNet | | ImageNet-1K | | |
|---|---|---|---|---|---|---|---|---|
| | | 10 | 50 | 10 | 50 | 10 | 50 | 100 |
| Hard Label | CE | 26.6 ± 0.4 | 40.8 ± 0.1 | 14.2 ± 0.3 | 26.9 ± 0.6 | 9.1 ± 0.1 | 17.5 ± 0.1 | 21.5 ± 0.1 |
| Soft Label | KL | 47.5 ± 0.3 | 55.7 ± 0.4 | 41.4 ± 0.3 | 47.2 ± 0.1 | 20.1 ± 0.4 | 38.5 ± 0.2 | 41.8 ± 0.2 |
| | JS | 47.9 ± 0.1 | 55.9 ± 0.3 | 41.8 ± 0.2 | 47.3 ± 0.2 | 20.5 ± 0.3 | 38.6 ± 0.3 | 41.9 ± 0.3 |
| | MSE | 47.6 ± 0.4 | 55.9 ± 0.1 | 41.6 ± 0.2 | 47.3 ± 0.0 | 20.7 ± 0.4 | 38.8 ± 0.4 | 41.9 ± 0.1 |
| | Soft CE | 40.8 ± 0.3 | 52.1 ± 0.3 | 33.4 ± 0.2 | 44.1 ± 0.4 | 17.0 ± 0.3 | 30.5 ± 0.8 | 37.2 ± 0.6 |
| Hard& Soft Label | KL + CE | 48.2 ± 0.4 | 56.3 ± 0.5 | 41.6± 0.5 | 46.8± 0.5 | 20.3 ± 0.2 | 35.2 ± 0.2 | 39.0 ± 0.2 |
| | MSE + CE | 47.4 ± 0.2 | 56.2 ± 0.0 | 41.7 ± 0.5 | 47.1 ± 0.1 | 20.5 ± 0.4 | 38.3 ± 0.3 | 40.5 ± 0.2 |
| | Soft CE + CE | 39.7 ± 0.3 | 51.1 ± 0.4 | 32.6 ± 0.5 | 43.2 ± 0.2 | 15.2 ± 0.4 | 29.1 ± 0.8 | 34.7 ± 0.6 |
| GIFT (ours) | | **50.6 ± 0.3** (↑ 2.4) | **57.9 ± 0.2** (↑ 1.6) | **44.0 ± 0.2** (↑ 2.3) | **48.3 ± 0.1** (↑ 1.0) | **24.0 ± 0.8** (↑ 3.3) | **39.5 ± 0.1** (↑ 0.7) | **42.5± 0.1** (↑ 0.6) |

SGD and Adam, exhibit distinct characteristics that influence model performance and generalization. Therefore, in practical applications, it is necessary to choose the most appropriate optimizer according to the training conditions. In this experiment, we report the accuracy before and after applying our GIFT to baseline methods across multiple optimizers not seen during dataset distillation, as shown in Table 8. The results clearly indicate that GIFT *enhances the generalization ability of all baseline methods.* More results on ImageNet-1K can be found in Table 16 in Appendix D.

It is notable that SRe$^2$L and RDED perform poorly in this cross-optimizer challenge. However, with our method, performance increased by 42.6% and 51.5% using SGD on CIFAR-100. We present an emperical and theoretical analysis of cross-optimizer generalization in Appendix E.

## 5.6 ABLATION STUDY

**Is Our Loss Function the Best?** To validate the superiority of GIFT, we compare with existing loss functions, utilizing both hard and soft labels. The results, presented in Table 9, clearly demonstrate that GIFT *consistently outperforms other loss functions and their combinations*.

For hard label utilization, the cross-entropy (CE) loss exhibits subpar performance, mainly due to the limited information content in typically small synthetic datasets. For soft label utilization, Jensen-Shannon (JS) divergence loss marginally outperforms the KL divergence, consistent with observations in (Kim et al., 2021). In summary, existing loss functions in synthetic datasets fail to fully exploit the potential of all labels. In contrast, our method leverages both hard and soft labels simultaneously, thereby maximizing label utilization potential and improving performance.

**Are Both Modules of GIFT Necessary?** We conduct an ablation study to assess the necessity of the label refinement and cosine similarity loss function on the small-scale dataset in Table 10 and the large-scale dataset ImageNet-1K in Table 17 in Appendix D. We compare the complete method with variants lacking either the teacher label refinement or the cosine similarity loss function. In the absence of both modules, the method is trained using its native loss function.

It is obvious that when only one module is employed, the cosine similarity loss function significantly enhances performance due to its direct label utilization. Label refinement consistently enhances performance, regardless of the presence of cosine similarity loss function, demonstrating its effectiveness. *Thus, both modules are essential for enhancement, consistent with our analysis in Section 4.*

Table 10: **Ablation study of label refinement (Refine) and cosine similarity loss function (Loss) on CIFAR 100 and Tiny-ImageNet when `IPC =10`.** This evaluation is conducted on both optimization-based (SRe$^2$L (Yin et al., 2023)) synthetic datasets and non-optimization-based (RDED (Sun et al., 2024)) synthetic datasets.

| DD Method | Refine | Loss | CIFAR100 | | | Tiny-ImageNet | | |
| --- | --- | --- | --- | --- | --- | --- | --- | --- |
| | | | 1 | 10 | 50 | 1 | 10 | 50 |
| SRe$^2$L | ✗ | ✗ | $13.6 \pm 0.4$ | $33.7 \pm 0.5$ | $52.3 \pm 0.2$ | $12.1 \pm 0.4$ | $34.5 \pm 0.4$ | $46.3 \pm 0.1$ |
| | ✓ | ✗ | $14.2 \pm 0.4$ (↑ 0.6) | $34.5 \pm 0.5$ (↑ 0.8) | $52.8 \pm 0.5$ (↑ 0.5) | $12.5 \pm 0.4$ (↑ 0.4) | $35.1 \pm 0.4$ (↑ 0.6) | $46.6 \pm 0.3$ (↑ 0.3) |
| | ✗ | ✓ | $14.7 \pm 0.4$ (↑ 1.1) | $37.3 \pm 0.4$ (↑ 3.6) | $54.6 \pm 0.1$ (↑ 2.3) | $12.7 \pm 0.4$ (↑ 0.6) | $36.9 \pm 0.3$ (↑ 2.4) | $46.9 \pm 0.1$ (↑ 0.6) |
| | ✓ | ✓ | $15.1 \pm 0.3$ (↑ 1.5) | $38.0 \pm 0.5$ (↑ 4.3) | $55.4 \pm 0.1$ (↑ 3.1) | $13.1 \pm 0.2$ (↑ 1.0) | $37.5 \pm 0.3$ (↑ 3.0) | $47.1 \pm 0.1$ (↑ 0.8) |
| RDED | ✗ | ✗ | $22.1 \pm 0.3$ | $47.5 \pm 0.3$ | $55.7 \pm 0.4$ | $17.9 \pm 0.3$ | $41.4 \pm 0.3$ | $47.2 \pm 0.1$ |
| | ✓ | ✗ | $22.9 \pm 0.3$ (↑ 0.8) | $48.0 \pm 0.3$ (↑ 0.5) | $56.3 \pm 0.1$ (↑ 0.6) | $18.2 \pm 0.3$ (↑ 0.3) | $41.9 \pm 0.4$ (↑ 0.5) | $48.1 \pm 0.2$ (↑ 0.9) |
| | ✗ | ✓ | $23.8 \pm 0.2$ (↑ 1.7) | $49.5 \pm 0.2$ (↑ 2.0) | $57.0 \pm 0.1$ (↑ 1.3) | $18.5 \pm 0.3$ (↑ 0.6) | $42.9 \pm 0.2$ (↑ 1.5) | $47.5 \pm 0.1$ (↑ 0.3) |
| | ✓ | ✓ | $24.7 \pm 0.3$ (↑ 2.6) | $50.6 \pm 0.3$ (↑ 3.1) | $57.9 \pm 0.2$ (↑ 2.2) | $19.1 \pm 0.3$ (↑ 1.2) | $44.0 \pm 0.2$ (↑ 2.6) | $48.3 \pm 0.1$ (↑ 1.1) |

Moreover, to verfiy the efficacy of label refinement, we compare the label accuracy before and after refinement. The results, shown in Figure 10 in Appendix D , demonstrate that it leads to significant performance improvements, highlighting the critical role of the proposed label refinement.

**Influence of Hyper-parameter $\gamma$.** We examine the impact of the weight hyperparameter $\gamma$, defined in Section 4 . As shown in Figure 2 , GIFT *achieves optimal performance when $\gamma = 0.1$ across various datasets*. This consistency is attributed to the fact that the soft labels are generated by pre-trained models. Specifically, both RDED and SRe$^2$L utilize the same pre-trained model.

Notably, for values of $\gamma$ greater than 0.1, a significant performance decline is observed across all methods as $\gamma$ increases. A plausible explanation is that larger values of $\gamma$ diminish the intra-class information content in soft labels. This observation aligns with our findings in Table 9 , where training with exclusively hard labels via CE results in poor performance. To verify that $\gamma = 0.1$ is also optimal for different settings, we conduct experiments on different network architectures and augmentation, as shown in Figure 8 and Figure 9 in Appendix D .

**Can GIFT Enhance utilization of Hard and Smoothed Labels?** *To evaluate the efficacy of GIFT across various data types*, we conduct experiments on both distilled and randomly selected datasets, employing hard and smoothed labels. The results are presented in Table 18 and Table 19 in Appendix D . Obviously, GIFT *consistently enhances label utilization across various label types*.

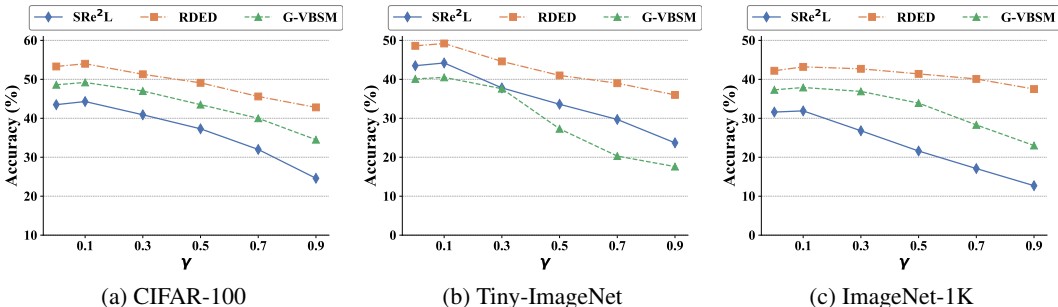

(a) CIFAR-100      (b) Tiny-ImageNet      (c) ImageNet-1K

Figure 2: Top-1 accuracy (%) for the state-of-the-art dataset distillation methods on various synthetic datasets when `IPC =10` on ResNet-18 with different $\gamma$.

## 6 CONCLUSION

This work introduces a novel perspective on dataset distillation by emphasizing the full utilization of synthetic labels. We first conduct a comprehensive comparison of existing loss functions used for soft labels in the field of dataset distillation. Our findings reveal that models trained on synthetic datasets exhibit significant sensitivity to the choice of loss function. Building on these observations, we propose a simple yet effective plug-and-play method, GIFT, which fully leverages synthetic labels without requiring additional information. The method incorporates label refinement and introduces a cosine similarity-based loss function. Furthermore, we provide a theoretical analysis to substantiate the use of cosine similarity. Experimental results across various scales and resolutions of image datasets demonstrate that GIFT consistently achieves superior performance compared to state-of-the-art dataset distillation methods.

ACKNOWLEDGMENT

We thank anonymous reviewers for their precious comments and feedback. This work was supported in part by the National Science and Technology Major Project (No. 2022ZD0115101), Research Center for Industries of the Future (RCIF) at Westlake University, Westlake Education Foundation, and Westlake University Center for High-performance Computing.

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

## A    PROOF OF THEOREM 1

*Proof.* The InfoNCE loss is defined as:

$$\mathcal{L}_{\text{InfoNCE}} = -\mathbb{E}\left[\log \frac{\exp(\text{sim}(z_i, y_i)/\tau)}{\sum_{j=1}^{N} \exp(\text{sim}(z_i, y_j)/\tau)}\right] \tag{4}$$

where $\text{sim}(z_i, y_i)$ represents the cosine similarity between $x_i$ and $y_i$:

$$\text{sim}(z_i, y_i) = \frac{z_i \cdot y_i}{\|z_i\|\|y_i\|} \tag{5}$$

Substituting the expression for cosine similarity into the InfoNCE loss:

$$
\begin{aligned}
\mathcal{L}_{\text{InfoNCE}} &= -\mathbb{E}\left[\log \frac{\exp\left(\frac{z_i \cdot y_i}{\|z_i\|\|y_i\|\tau}\right)}{\sum_{j=1}^{N} \exp\left(\frac{z_i \cdot y_j}{\|z_i\|\|y_j\|\tau}\right)}\right] \\
&= -\mathbb{E}\left[\log \exp\left(\frac{z_i \cdot y_i}{\|z_i\|\|y_i\|\tau}\right) - \log \sum_{j=1}^{N} \exp\left(\frac{z_i \cdot y_j}{\|z_i\|\|y_j\|\tau}\right)\right] \\
&= -\mathbb{E}\left[\left(\frac{z_i \cdot y_i}{\|z_i\|\|y_i\|\tau}\right) - \log \sum_{j=1}^{N} \exp\left(\frac{z_i \cdot y_j}{\|z_i\|\|y_j\|\tau}\right)\right] \\
&= -\mathbb{E}\left[\left(\frac{z_i \cdot y_i}{\|z_i\|\|y_i\|\tau}\right)\right] + \mathbb{E}\left[\log \sum_{j=1}^{N} \exp\left(\frac{z_i \cdot y_j}{\|z_i\|\|y_j\|\tau}\right)\right]
\end{aligned}
\tag{6}
$$

Applying Jensen's inequality to the logarithm:

$$\mathcal{L}_{\text{InfoNCE}} \leq -\mathbb{E}\left[\left(\frac{z_i \cdot y_i}{\|z_i\|\|y_i\|\tau}\right)\right] + \log\left(\mathbb{E}\left[\sum_{j=1}^{N} \exp\left(\frac{z_i \cdot y_j}{\|z_i\|\|y_j\|\tau}\right)\right]\right) \tag{7}$$

Assuming the negative samples $y_j$ are drawn from a similar distribution, we approximate the denominator:

$$\sum_{j=1}^{N} \exp\left(\frac{z_i \cdot y_j}{\|z_i\|\|y_j\|\tau}\right) \approx N \exp\left(\frac{\mathbb{E}[z_i \cdot y_j]}{\|z_i\|\mathbb{E}[\|y_j\|]\tau}\right) \tag{8}$$

Substituting this approximation into the upper bound:

$$\boxed{\mathcal{L}_{\text{InfoNCE}} \leq -\frac{1}{\tau}\left(\mathbb{E}\left[\left(\frac{z_i \cdot y_i}{\|z_i\|\|y_i\|}\right)\right] - \frac{\mathbb{E}[z_i \cdot y_j]}{\|z_i\|\mathbb{E}[\|y_j\|]}\right) + \log(N)} \tag{9}$$

$\square$

## B    RELATED WORK

### B.1    KNOWLEDGE DISTILLATION

A straightforward method to simultaneously utilize soft and hard labels is knowledge distillation (Hinton et al., 2015), which transfers knowledge from a large teacher model to a small student model. In this training process, the student model is supervised by hard labels and soft labels from the teacher's output. Many following works aim to enhance the use of soft labels for more effective knowledge transfer. (Yuan et al., 2020) investigated the regularization property of soft labels and introduced a teacher-free distillation approach. WSLD (Zhou et al., 2021) analyzes soft labels and distributes different weights for them from a perspective of bias-variance trade-off. DKD (Zhao et al., 2022) decouples the logit and assigns different weights for the target and non-target classes.

Despite the promising potential of knowledge distillation in transferring knowledge from teacher to student models using soft labels, its application to our problem yields limited improvement. A detailed analysis and comparison of these limitations are provided in Section 5.3 .

## C    EXPERIMENT DETAILS

**Datasets.**    As described in Section 5.1, we evaluate the state-of-the-art dataset distillation methods and our proposed GIFT on both small-scale and large-scale datasets. More Information about datasets utilized are listed in Table 11.

Table 11: Details about the datasets

| Dataset | Num of Classes | IPC of Trainset | IPC of Testset |
|---|---|---|---|
| CIFAR-10 | 10 | 5000 | 1000 |
| CIFAR-100 | 100 | 500 | 100 |
| Tiny-ImageNet | 200 | 500 | 50 |
| ImageNet-1k | 1000 | 732 - 1300 | 50 |

**Models.**    The experiment utilized a plethora of pre-trained models, and we provided the accuracy of these pre-trained models in the Table 12. The results are provided for reference only.

**Baselines.**    To elucidate the rationale behind our method selection for comparison, we categorize current state-of-the-art methods based on two key factors: scalability to ImageNet-1K and the utilization of soft labels. These categorizations are summarized in Table 13.

Given that our primary focus is on enhancing the use of soft labels in dataset distillation, we restrict our comparisons to methods that involve soft labels:

- TESLA (Cui et al., 2023) marks the first distillation approach that *extends to the full ImageNet-1K*, circumventing the extensive memory demands associated with MTT-derived methods through a constant memory footprint. However, TESLA does not provide public synthesic datasets, so we are not able to conduct on it.
- SRe$^2$L (Yin et al., 2023) and RDED (Sun et al., 2024): both of them use soft labels assigned by a teacher model and use them via KL divergence.
- DATM (Guo et al., 2024): initial soft labels assigned by multiple teacher models and then optimized based on trajectory matching. Finally, this method employs soft cross-entropy loss for soft labels.
- G-VBSM (Shao et al., 2024): soft labels are assigned by multiple teacher models and then used via MSE-CE loss function.
- CDA (Yin & Shen, 2023): soft labels are assigned by a teacher model and are used via soft cross-entropy loss.

Knowledge distillation (Hinton et al., 2015) is a straightforward method to utilize labels, especially for soft labels. Therefore, we also compare our method GIFT with the state-of-the-art knowledge distillation methods that focus on soft labels utilization:

Table 12: Accuracy of pretrained models.

| Dataset | Model | Size | Accuracy |
|---|---|---|---|
| CIFAR-10 | resnet18_modified | $32 \times 32$ | 93.86 |
|  | ConvNet-3 | $32 \times 32$ | 82.24 |
| CIFAR-100 | resnet18_modified | $32 \times 32$ | 72.27 |
|  | ConvNet-3 | $32 \times 32$ | 61.27 |
| Tiny-ImageNet | resnet18_modified | $64 \times 64$ | 61.98 |
|  | ConvNet-4 | $64 \times 64$ | 49.73 |
| ImageNet-1k | resnet18 | $224 \times 224$ | 69.31 |
|  | ConvNet-4 | $64 \times 64$ | 43.6 |

Table 13: **Categorize methods based on their utilization of soft labels and their scalability to ImageNet-1K.**

| | RDED | CDA | G-VBSM | SRe2L | DATM | SeqMatch | DREAM | IDC | FTD | DataDAM | MTT | DM | DSA |
|---|---|---|---|---|---|---|---|---|---|---|---|---|---|
| Use Soft Label | ✓ | ✓ | ✓ | ✓ | ✓ | ✗ | ✗ | ✗ | ✗ | ✗ | ✗ | ✗ | ✗ |
| Scale to ImageNet-1K | ✓ | ✓ | ✓ | ✓ | ✗ | ✗ | ✓ | ✗ | ✗ | ✗ | ✗ | ✗ | ✗ |

- KD (Hinton et al., 2015): it is the first method to transfer knowledge using both hard and soft labels from the teacher's output.
- WSLD (Zhou et al., 2021): it analyzes soft labels and then distributes different weights for them from a perspective of bias-variance trade-off.
- DKD (Zhao et al., 2022): it decouples the logits and assigns different weights for the target and non-target classes.
- NKD (Yang et al., 2023): it finds the sum of the two non-target logits is different, preventing logits' distributions from being identical. Therefore, it normalizes the non-target logits to equalize their sum.

**Evaluating main results.** For both dataset distillation and performance evaluation, we employ identical neural network architectures. Consistent with previous studies (Cazenavette et al., 2022; Cui et al., 2023; Zhao et al., 2023), we use Conv-3 for CIFAR-10 and CIFAR-100 distillation tasks, Conv-4 for Tiny-ImageNet (with the exception of DREAM, which utilizes Conv-3) and ImageNet-1K, Conv-5 for ImageNet-10, and Conv-6 for ImageNet-100 distillation. In line with (Cazenavette et al., 2022; Cui et al., 2023), MTT and TESLA apply a reduced resolution for distilling $224 \times 224$ images. According to (Yin et al., 2023), for retrieving and evaluating distilled datasets, SRe$^2$L and GIFT adopt ResNet-18.

**Evaluating the distilled dataset.** Consistent with recent works (Yin et al., 2023; Shao et al., 2024), the evaluation on the distilled dataset follows the parameters outlined in Table 15 . Furthermore, we implement Differentiable Siamese Augmentation (DSA) as described by (Zhao & Bilen, 2021) to enhance images during both the distillation and evaluation phases of our experiments.

**Differentiable Siamese Augmentation (DSA).** We use DSA (Differentiable Siamese Augmentation) as a tool for image augmentation. For the sake of clarity, we delineate the DSA operations utilized in Table 14 , alongside their respective transforms and probabilities.

Table 14: Differentiable Siamese Augmentation(DSA) and ratios

| DSA | Transform | Ratio |
|---|---|---|
| Color | Color Jitter | Brightness=1.0 Saturation=2.0 Contrast=0.5 |
| Crop | Random Crop | Crop Pad=0.125 |
| Cutout | Random Cutout | Cutout=0.5 |
| Flip | Random Horizontal Flip | Flip=0.5 |
| Scale | Random Scale | Scale=1.2 |
| Rotate | Random Rotation | Rotate=15.0 |

## D    EXPERIMENT RESULTS

**Application: Continual Learning** Following prior studies (Zhao & Bilen, 2023; Kim et al., 2022; Yin et al., 2023) that leverage synthetic datasets in continual learning to assess the quality of synthetic data, we employ the GDumb framework (Prabhu et al., 2020) for our continual learning setup. This framework sequentially stores prior training data in memory and utilizes both new and stored data for model training.

We conduct class-incremental learning on Tiny-ImageNet with an `IPC` =10 using ResNet-18. Figure 3 illustrates both 5-step and 10-step class-incremental learning strategies, partitioning the 200

Table 15: Evaluation Hyperparameter setting

| Config | Value | Explanation |
|---|---|---|
| Epochs | 300/1000 | 300 for ImageNet-1k,
1000 for default |
| Optimizer
Learning Rate | AdamW
0.001 | NA
NA |
| Batch Size | 10/50/100/200 | 10 for $0 <$ Num of Images $\leq 10$,
50 for $10 <$ Num of Images $\leq 500$,
100 for $500 <$ Num of Images $\leq 20000$,
200 for $20000 <$ Num of Images |
| Scheduler | MultiStepLR | milestones=[2 × epochs // 3, 5 × epochs // 6]
gamma=0.2 |
| Augmentation | DSA strategy | color, crop, cutout, flip, scale, rotate |

Table 16: **Top-1 accuracy on cross-optimization generalization on ImageNet-1K when IPC=10.** We evaluate the performance of synthetic datasets across various optimizers.

| | ImageNet-1K | | | | | |
|---|---|---|---|---|---|---|
| Dataset | ConvNet | | | ResNet | | |
| Method | SGD | Adam | AdamW | SGD | Adam | AdamW |
| SRe$^2$L | 0.1 ± 0.0 | 0.1 ± 0.0 | 12.5 ± 0.1 | 0.1 ± 0. | 0.1 ± 0.0 | 31.5 ± 0.3 |
| SRe$^2$L + Ours | 18.2 ± 0.2 (↑ 18.1) | 26.6 ± 0.2 (↑ 26.5) | 14.2 ± 0.6 (↑ 1.7) | 36.1 ± 0.1 (↑ 36.0) | 24.5 ± 0.2 (↑ 24.4) | 31.9 ± 0.2 (↑ 0.4) |
| RDED | 0.1 ± 0.0 | 0.1 ± 0.0 | 20.1 ± 0.4 | 0.1 ± 0.0 | 0.1 ± 0.0 | 41.4 ± 0.4 |
| RDED + Ours | 26.7 ± 0.6 (↑ 26.5) | 30.9 ± 0.7 (↑ 26.5) | 24.0 ± 0.8 (↑ 3.2) | 45.8 ± 0.4 (↑ 30.8) | 29.1 ± 0.3 (↑ 29.0) | 43.2 ± 0.1 (↑ 1.8) |
| G-VBSM | 20.4 ± 0.8 | 25.0 ± 0.4 | 22.6 ± 0.5 | 38.7 ± 0.2 | 27.0 ± 0.2 | 36.7 ± 0.2 |
| G-VBSM + Ours | 27.4 ± 0.8 (↑ 7.0) | 29.8 ± 0.5 (↑ 4.8) | 24.3 ± 0.2 (↑ 1.7) | 41.8 ± 0.1 (↑ 3.1) | 27.8 ± 0.8 (↑ 0.8) | 37.9 ± 0.5 (↑ 1.2) |

classes into either 5 or 10 learning steps, corresponding to 40 and 20 classes per step, respectively. It is evident that *our results substantially improve upon the baseline method RDED* [7].

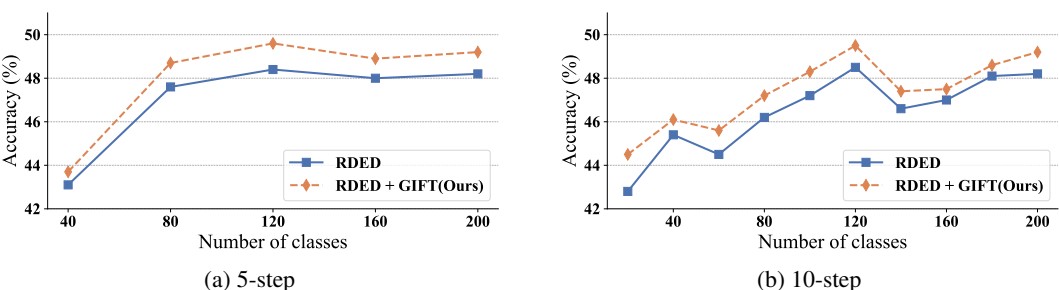

(a) 5-step

(b) 10-step

Figure 3: **5-step and 10-step class-incremental learning on Tiny-ImageNet on ResNet-18**.

**Comprehensive Comparison Between Different Loss Functions.** Our experiments span two datasets, including Tiny-ImageNet, and ImageNet-1K on IPC $\in \{1, 10, 50\}$ using ConvNet (Guo et al., 2024). Note that the synthetic dataset for DATM on ImageNet-1K is unavailable, precluding comparisons on this dataset. We evaluate at IPC $\in \{1, 10, 50\}$. The results, visualized in Figure 4, Figure 6, Figure 5 and Figure 7, reveal that *the performance of models trained on synthetic datasets is highly sensitive to the choice of the loss function*, highlighting the necessity for a unified and effective loss function in dataset distillation.

**Influence of Hyper-parameter $\gamma$ on ConvNet.** We also examined the impact of the hyper-parameter $\gamma$ using ConvNet, and the results are shown in Figure 8. Similar to the findings with ResNet, GIFT *achieves optimal performance when $\gamma = 0.1$.*

---

[7]RDED is the current state-of-the-art method as shown in Table 1, so we conduct the experiment on its synthetic datasets.

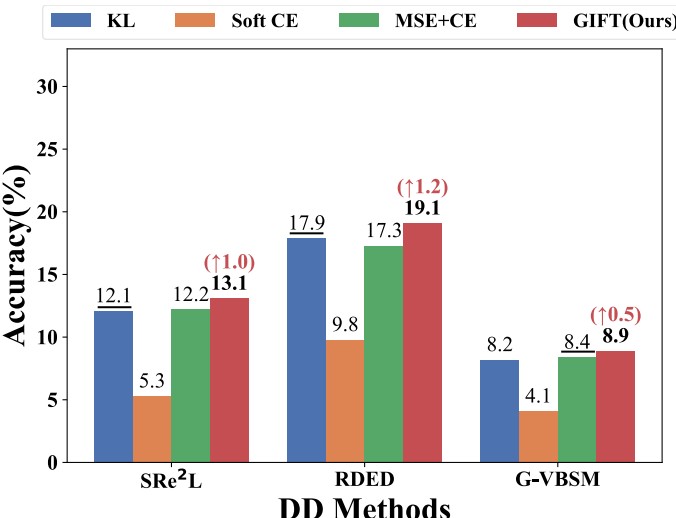

Figure 4: Top-1 accuracy on various synthetic datasets via the SOTA dataset distillation methods across loss functions on Tiny-ImageNet when IPC=1.

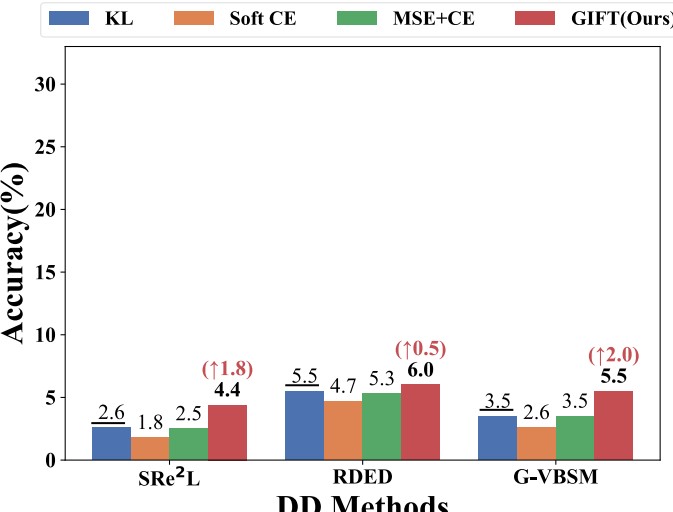

Figure 5: Top-1 accuracy on various synthetic datasets via the SOTA dataset distillation methods across loss functions on ImageNet-1K when IPC=1.

**Ablation study on ImageNet-1K.** We also conducted an ablation study to assess the necessity of the teacher label refinement and cosine similarity loss function on the large-scale dataset in Table 17 . This evaluation was performed on both optimization-based (SRe$^2$L) and non-optimization-based (RDED) synthetic datasets. It is obvious *both modules are essential for performance enhancement, consistent with our analysis in Section 4 .*

**Efficacy of Label Refinement.** The necessity of incorporating hard labels into the soft labels generated by teacher models arises from the inherent limitations in the performance of these teacher models. Specifically, the test accuracies of teacher models are only 61.27%, 49.73%, and 43.6% for CIFAR-100, Tiny-ImageNet, and ImageNet-1K, respectively, when trained on commonly used ConvNet architectures in dataset distillation. Consequently, the accuracy of soft labels is constrained by the suboptimal nature of the teacher models. To mitigate potential inaccuracies in these soft labels, we integrate hard labels to enhance reliability.

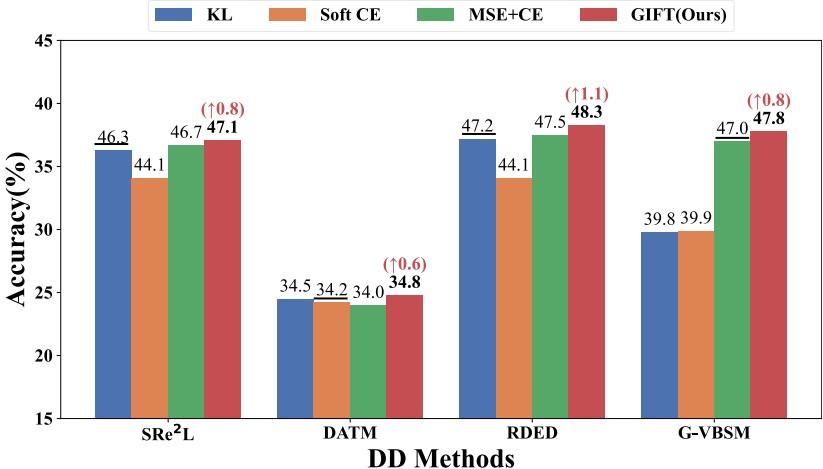

Figure 6: Top-1 accuracy on various synthetic datasets via the SOTA dataset distillation methods across loss functions on Tiny-ImageNet when IPC=50.

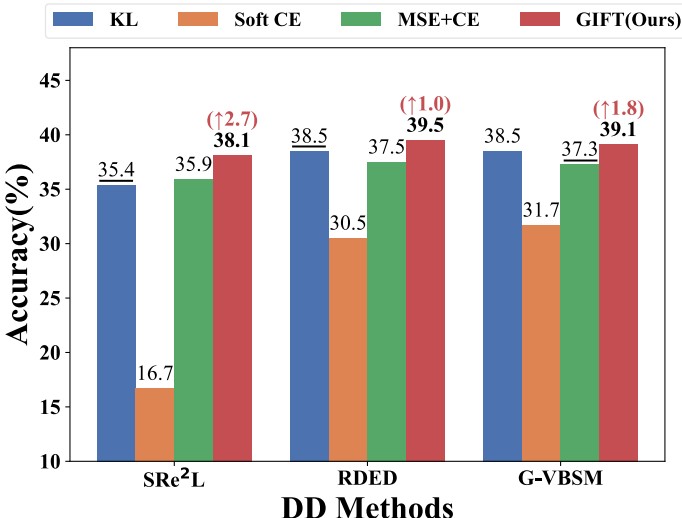

Figure 7: Top-1 accuracy on various synthetic datasets via the SOTA dataset distillation methods across loss functions on ImageNet-1K when IPC=50.

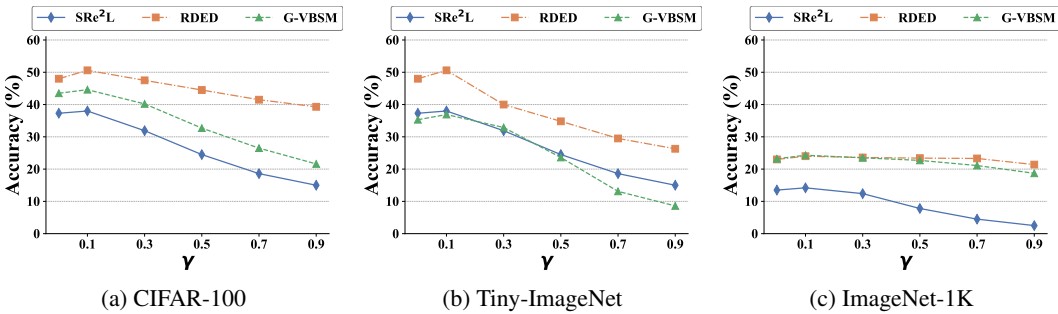

Figure 8: Top-1 accuracy for the SOTA dataset distillation methods on various synthetic datasets when `IPC` =10 on ConvNet with different $\gamma$.

To verify the efficacy of the label refinement, we record the label accuracy before and after refinement across each training epoch for three datasets. The parameter $\gamma$, controlling the integration ratio, is set

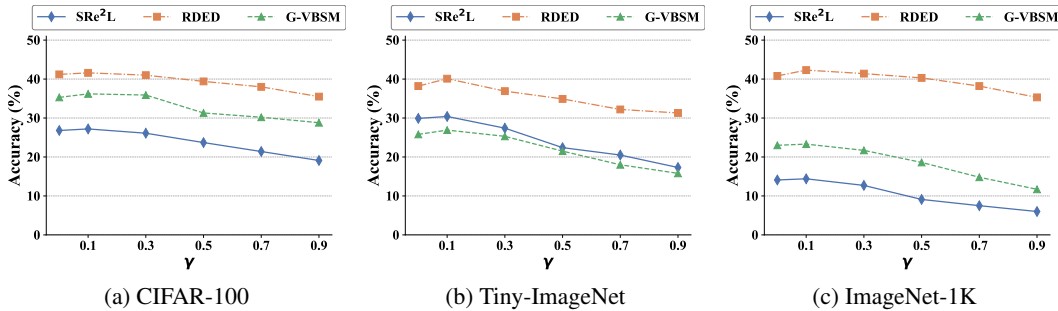

Figure 9: Top-1 accuracy for the SOTA dataset distillation methods on various synthetic datasets with different data augmentation techniques when `IPC` =10 on ResNet-18 with different $\gamma$.

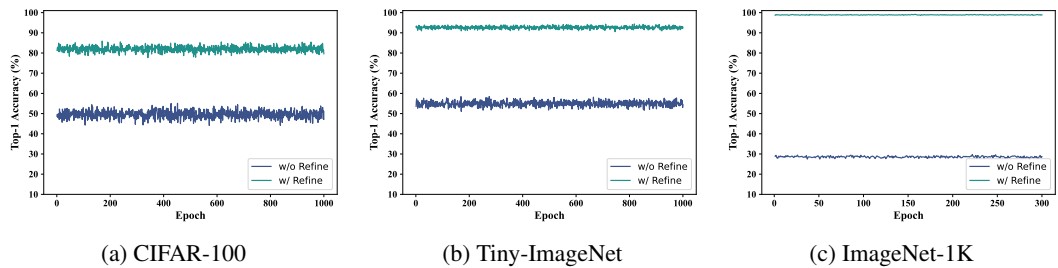

Figure 10: Top-1 accuracy for the SOTA dataset distillation methods on various synthetic datasets when `IPC` =10 on ResNet-18.

Table 17: **Ablation study of label refinement (Refine) and cosine similarity loss function (Loss) on ImageNet-1K.** In the table, (↑) means the improvements over these methods.

| | ImageNet-1K | | | | | |
| | ConvNet | | | ResNet-18 | | |
| Method | 10 | 50 | 100 | 10 | 50 | 100 |
|---|---|---|---|---|---|---|
| SRe$^2$L | 12.5 ± 0.3 | 35.4 ± 1.0 | 40.1 ± 0.4 | 31.5 ± 0.3 | 49.5 ± 0.1 | 54.3 ± 0.2 |
| SRe$^2$L + Refine | 13.3 ± 0.4 (↑ 0.8) | 36.3 ± 0.4 (↑ 0.9) | 40.5 ± 0.4 (↑ 0.4) | 31.8 ± 0.3 (↑ 0.3) | 49.7 ± 0.3 (↑ 0.2) | 54.5 ± 0.1 (↑ 0.2) |
| SRe$^2$L + Loss | 13.1 ± 0.6 (↑ 0.6) | 36.8 ± 0.3 (↑ 1.2) | 40.7 ± 0.3 (↑ 0.6) | 31.7 ± 0.2 (↑ 0.2) | 49.8 ± 0.2 (↑ 0.3) | 54.5 ± 0.2 (↑ 0.2) |
| SRe$^2$L + Refine + Loss | 14.2 ± 0.6 (↑ 1.7) | 38.1 ± 0.4 (↑ 2.7) | 41.5 ± 0.2 (↑ 1.4) | 31.9 ± 0.1 (↑ 0.4) | 50.1 ± 0.2 (↑ 0.6) | 54.8 ± 0.1 (↑ 0.5) |
| RDED | 20.1 ± 0.4 | 38.5 ± 0.2 | 41.8 ± 0.2 | 41.4 ± 0.4 | 55.5 ± 0.2 | 58.8 ± 0.1 |
| RDED + Refine | 21.2 ± 0.5 (↑ 1.2) | 38.7 ± 0.2 (↑ 0.2) | 42.3 ± 0.4 (↑ 0.5) | 42.3 ± 0.2 (↑ 0.9) | 55.8 ± 0.3 (↑ 0.3) | 59.2 ± 0.3 (↑ 0.4) |
| RDED + Loss | 21.7 ± 0.6 (↑ 1.6) | 39.3 ± 0.1 (↑ 0.8) | 42.1 ± 0.1 (↑ 0.3) | 42.0 ± 0.2 (↑ 0.6) | 55.9 ± 0.1 (↑ 0.4) | 59.1 ± 0.1 (↑ 0.3) |
| RDED + Refine + Loss | 24.0 ± 0.8 (↑ 3.9) | 39.5 ± 0.1 (↑ 1.0) | 42.5 ± 0.1 (↑ 0.7) | 43.2 ± 0.1 (↑ 1.8) | 56.5 ± 0.2 (↑ 1.0) | 59.3 ± 0.1 (↑ 0.5) |
| G-VBSM | 22.6 ± 0.5 | 37.3 ± 0.3 | 40.1 ± 0.4 | 36.7 ± 0.2 | 52.3 ± 0.1 | 57.3 ± 0.1 |
| G-VBSM + Refine | 23.3 ± 0.4 (↑ 0.7) | 37.8 ± 0.3 (↑ 0.5) | 40.9 ± 0.3 (↑ 0.7) | 37.2 ± 0.3 (↑ 0.5) | 52.7 ± 0.2 (↑ 0.4) | 57.5 ± 0.2 (↑ 0.2) |
| G-VBSM + Loss | 23.8 ± 0.2 (↑ 1.2) | 38.3 ± 0.4 (↑ 1.0) | 41.8 ± 0.3 (↑ 1.7) | 37.5 ± 0.5 (↑ 0.8) | 52.8 ± 0.2 (↑ 0.5) | 57.4 ± 0.1 (↑ 0.1) |
| G-VBSM + Refine + Loss | 24.3 ± 0.2 (↑ 1.7) | 39.1 ± 0.3 (↑ 1.8) | 42.1 ± 0.3 (↑ 2.0) | 37.9 ± 0.5 (↑ 1.2) | 53.1 ± 0.2 (↑ 0.8) | 57.6 ± 0.1 (↑ 0.3) |

to 0.1. As depicted in Figure 10, refining soft labels results in significant performance improvements of 37.1%, 40.95%, and 71.39% for CIFAR-100, Tiny-ImageNet, and ImageNet-1K, respectively, highlighting the critical role of the proposed label refinement.

**Cross-Optimizaer on ImageNet-1K** In this experiment, we report the accuracy before and after applying our GIFT to baseline methods across multiple optimizers not seen during dataset distillation onImageNet-1K , as shown in Table 16. The results clearly indicate that GIFT *enhances the generalization ability of all baseline methods.* This leads to more stable gradients, especially in scenarios with small dataset sizes.

**GIFT Enhance Utilization of Hard and Smoothed Labels** To evaluate the efficacy of GIFT across various data types, we conduct experiments on both distilled and randomly selected datasets, employing hard and smoothed labels, with `IPC` =10. (1) For the distilled dataset, we use the state-of-the-art dataset distillation method, RDED, which uses soft labels generated by teacher models. In the experiments, we maintain the distilled images and replace soft labels with hard and smoothed

labels for model training. (2) For random dataset, we randomly select 10 images for each class from the original dataset. The results for two types of data are presented in Table 18 and Table 19 . It is evident that *applying our method to label utilization consistently improves performance for both the distilled and the randomly selected data.*

Table 18: Evaluation of Loss Functions Across Hard and Smoothed Labels on the Distilled Data Generated via RDED with `IPC` =10.

| Label Type | Loss Function | CIFAR100 | Tiny-ImageNet | ImageNet-1K |
|---|---|---|---|---|
| Hard | CE | $21.6 \pm 0.2$ | $13.5 \pm 0.3$ | $8.3 \pm 0.4$ |
| | Ours | $22.8 \pm 0.2$ (↑ 1.2) | $14.8 \pm 0.3$ (↑ 1.3) | $9.8 \pm 0.2$ (↑ 1.5) |
| Smoothed | SoftCE | $21.9 \pm 0.2$ | $13.8 \pm 0.2$ | $8.5 \pm 0.2$ |
| | KL | $21.7 \pm 0.3$ | $13.3 \pm 0.3$ | $8.0 \pm 0.3$ |
| | MSE | $22.1 \pm 0.1$ | $14.1 \pm 0.1$ | $8.8 \pm 0.3$ |
| | Ours | $23.1 \pm 0.2$ (↑ 1.0) | $15.2 \pm 0.3$ (↑ 1.1) | $10.2 \pm 0.2$ (↑ 1.4) |

Table 19: Evaluation of Loss Functions Across Hard and Smoothed Labels on the Randomly Selected Data with IPC=10.

| Label Type | Loss Function | CIFAR100 | Tiny-ImageNet | ImageNet-1K |
|---|---|---|---|---|
| Hard | CE | $18.9 \pm 0.3$ | $9.4 \pm 0.1$ | $4.0 \pm 0.3$ |
| | Ours | $20.3 \pm 0.1$ (↑ 1.4) | $11.6 \pm 0.1$ (↑ 2.2) | $5.1 \pm 0.2$ (↑ 1.1) |
| Smoothed | SoftCE | $19.0 \pm 0.3$ | $9.8 \pm 0.3$ | $4.5 \pm 0.2$ |
| | KL | $17.9 \pm 0.2$ | $8.5 \pm 0.3$ | $3.9 \pm 0.1$ |
| | MSE | $19.2 \pm 0.1$ | $9.7 \pm 0.2$ | $4.6 \pm 0.3$ |
| | Ours | $20.4 \pm 0.2$ (↑ 1.2) | $11.8 \pm 0.2$ (↑ 2.0) | $5.6 \pm 0.2$ (↑ 1.0) |

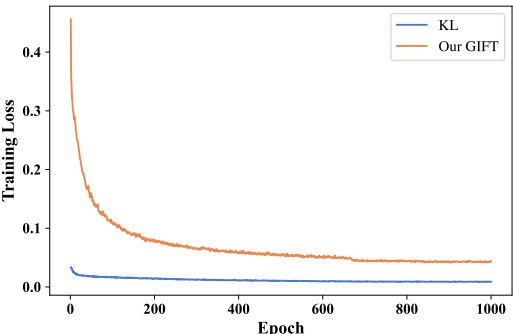

Figure 11: Comparison of Training Loss Between KL and Our GIFT Across Training Epochs

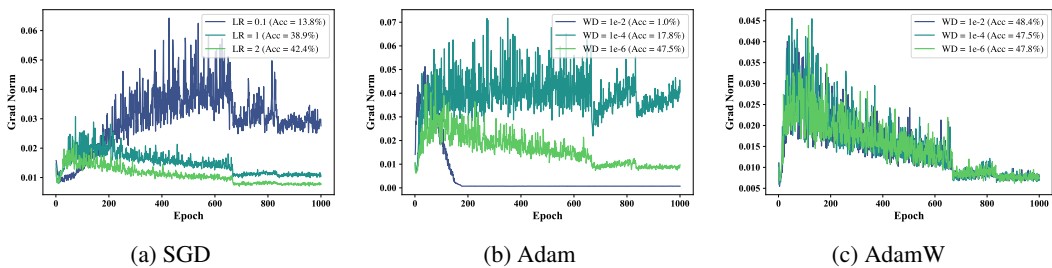

(a) SGD          (b) Adam          (c) AdamW

Figure 12: Gradient Norms Across Various Optimizers and Hyperparameters.

# E    EMPERICAL AND THEORETICAL ANALYSIS OF CROSS-OPTIMIZER GENERALIZATION

We begin by analyzing the three optimizers, namely:

- SGD (Robbins & Monro, 1951) directly computes gradients from loss values, significantly impacting the update of model parameters.
- Adam (Kingma & Ba, 2014) includes weight decay in the gradient computation, meaning that *when the primary gradient signal (from the loss) is small, weight decay can overshadow it, leading to ineffective updates toward minimizing the loss.
- AdamW (Loshchilov, 2017) separates the concerns of optimization and regularization. By applying weight decay independently, it ensures that the optimization process remains focused on minimizing the loss function, while regularization acts as a controlled adjustment to the parameter magnitudes.

Optimizers inherently exhibit diverse characteristics, resulting in distinct training dynamics when applied to distilled datasets generated by various methods. Specifically, *these distilled datasets, trained with varying loss functions, exhibit distinct loss values.* We reveal that the performance of the optimizers is **highly influenced** by these loss values, as demonstrated by the subsequent empirical evidence and theoretical analysis.

## E.1    EMPERICAL ANALYSIS

When employing the KL divergence loss function, the RDED generally exhibits low loss values, as depicted in Figure 11 . Notably, our GIFT framework does not achieve small loss values.

To examine the impact of loss values on optimizer performance, we conduct experiments utilizing RDED-generated distilled data. We train models using three distinct optimizers, each with distinct hyperparameter configurations, and compute the gradient norm for each. Specifically, we varied the learning rate for the SGD optimizer and modified the weight_decay parameter for both the Adam and AdamW optimizers.

The gradient norms of these models, presented in Figure 12 , demonstrate that when loss values are small, the training dynamics exhibit heightened sensitivity to the optimizer choice. Therefore, the performances of different optimizers are highly influenced by the loss values. Notably, our GIFT framework can not obtain small loss values, achieving robust performance across various optimizers.

## E.2    THEORETICAL ANALYSIS

### E.2.1    STOCHASTIC GRADIENT DESCENT (SGD)

Stochastic Gradient Descent (SGD) is a foundational optimization algorithm widely used for training machine learning models, particularly neural networks. SGD iteratively updates model parameters to minimize the loss function by moving in the direction of the negative gradient of the loss with respect to the parameters.

**SGD Update Rules.**    Define the following:

- $\theta_t$: Parameters at time step $t$.
- $g_t = \nabla_\theta L(\theta_{t-1})$: Gradient of the loss function $L$ with respect to parameters $\theta$ at time step $t - 1$.
- $\eta$: Learning rate.

The basic SGD update rule is:
$$\theta_t = \theta_{t-1} - \eta g_t$$

**Sensitivity to Loss Values in SGD.**    The gradient $g_t = \nabla_\theta L(\theta_{t-1})$ indicates the direction and magnitude of change needed to minimize the loss function. Larger loss function values typically

result in larger gradients. Therefore, in SGD, both the loss values and the learning rate $\eta$ directly affect the model updates.

### E.2.2 ADAM OPTIMIZER

Adam (Adaptive Moment Estimation) is an optimization algorithm that combines the advantages of two extensions of stochastic gradient descent: Adaptive Gradient Algorithm (AdaGrad) and Root Mean Square Propagation (RMSProp). Adam maintains per-parameter learning rates adapted based on the first and second moments of the gradients.

**Adam Update Rules.**   Define the following:

- $\theta_t$: Parameters at time step $t$.
- $g_t = \nabla_\theta L(\theta_{t-1})$: Gradient of the loss function $L$ with respect to parameters $\theta$ at time step $t - 1$.
- $m_t$: First moment estimate (exponentially decaying average of past gradients).
- $v_t$: Second moment estimate (exponentially decaying average of past squared gradients).
- $\beta_1, \beta_2$: Decay rates for the first and second moments, respectively.
- $\epsilon$: Small constant to prevent division by zero.
- $\eta$: Learning rate.

The update rules are as follows:

$$
\begin{aligned}
m_t &= \beta_1 m_{t-1} + (1 - \beta_1) g_t \\
v_t &= \beta_2 v_{t-1} + (1 - \beta_2) g_t^2 \\
\hat{m}_t &= \frac{m_t}{1 - \beta_1^t} && \text{(Bias-corrected first moment)} \\
\hat{v}_t &= \frac{v_t}{1 - \beta_2^t} && \text{(Bias-corrected second moment)} \\
\theta_t &= \theta_{t-1} - \eta \frac{\hat{m}_t}{\sqrt{\hat{v}_t} + \epsilon}
\end{aligned}
$$

**Weight Decay in Adam.**   In the original Adam implementation, weight decay is typically incorporated by adding an $L_2$ regularization term directly to the loss function:

$$
L'(\theta) = L(\theta) + \frac{\lambda}{2} \|\theta\|_2^2
$$

The gradient of the modified loss function with respect to $\theta$ is:

$$
\nabla_\theta L'(\theta) = \nabla_\theta L(\theta) + \lambda \theta
$$

Consequently, the gradient used in the Adam update rule is augmented with the weight decay term:

$$
g_t = \nabla_\theta L(\theta_{t-1}) + \lambda \theta_{t-1}
$$

Substituting this into the Adam update equations, the parameter update rule becomes:

$$
\theta_t = \theta_{t-1} - \eta \frac{\hat{m}_t}{\sqrt{\hat{v}_t} + \epsilon}
$$

Here, $\hat{m}_t$ and $\hat{v}_t$ incorporate the additional gradient component $\lambda \theta_{t-1}$.

### E.2.3 ADAMW OPTIMIZER

AdamW is a modification of the Adam optimizer that decouples weight decay from the gradient-based update. AdamW addresses the intertwined nature of weight decay and gradient updates in the original Adam, leading to improved generalization performance and more stable training dynamics.

**AdamW Update Rules.** The primary distinction in AdamW lies in how weight decay is applied. Instead of incorporating weight decay into the gradient computation, AdamW applies it directly to the parameters after the standard Adam update. The update rule is expressed as:

$$\theta_t = \theta_{t-1} - \eta \frac{\hat{m}_t}{\sqrt{\hat{v}_t} + \epsilon} - \eta \lambda \theta_{t-1}$$

Alternatively, this can be broken down into two sequential steps:

$$\theta_t' = \theta_{t-1} - \eta \frac{\hat{m}_t}{\sqrt{\hat{v}_t} + \epsilon}$$
$$\theta_t = \theta_t' - \eta \lambda \theta_{t-1}$$

In this formulation:

- The first step performs the standard Adam gradient-based update. - The second step applies
- weight decay independently of the gradient computation.

**Sensitivity to Loss Values in AdamW.** By decoupling weight decay from the gradient-based update, AdamW mitigates the issue of weight decay dominating parameter updates when the loss $L(\theta)$ is small. In AdamW, the gradient-based update remains primarily responsible for minimizing the loss, while weight decay independently enforces regularization. This separation ensures that even when $\nabla_\theta L(\theta)$ is minimal, the optimizer can continue to adjust parameters based on the loss gradient without being overly constrained by the weight decay term.

When $L(\theta_{t-1})$ is small, $\nabla_\theta L(\theta_{t-1}) \approx 0$. In AdamW, the update rule is:

$$\theta_t = \theta_{t-1} - \eta \frac{\hat{m}_t}{\sqrt{\hat{v}_t} + \epsilon} - \eta \lambda \theta_{t-1}$$

The gradient-based update $-\eta \frac{\hat{m}_t}{\sqrt{\hat{v}_t}+\epsilon}$ remains tied to the loss gradient, allowing continued optimization of $L(\theta)$. Simultaneously, the weight decay term $-\eta \lambda \theta_{t-1}$ independently controls the magnitude of $\theta$ without influencing the direction dictated by the loss gradient. This ensures that weight decay does not overshadow the gradient-based updates, enabling effective model training even when the loss is minimal.

### E.2.4 WHY EXCESSIVE WEIGHT DECAY IN ADAM IMPEDES UPDATES WHEN LOSS IS SMALL?

**Adam's Update Mechanism Under Small Loss.** Consider the Adam update rule with weight decay integrated into the gradient:

$$\theta_{t+1} = \theta_t - \eta \frac{\hat{m}_t}{\sqrt{\hat{v}_t} + \epsilon},$$

where $g_t = \nabla_\theta L(\theta_t) + \lambda \theta_t$ Assume that the loss $L(\theta_t)$ is sufficiently small, implying $\nabla_\theta L(\theta_t) \approx 0$. Thus $g_t \approx \lambda \theta_t$. Therefore, the update rule is expressed as:

$$\theta_{t+1} \approx \theta_t - \eta \frac{\lambda \theta_t}{\sqrt{v_t} + \epsilon} \approx \theta_t \left(1 - \frac{\eta \lambda}{\sqrt{v_t} + \epsilon}\right),$$

Here, the parameter $\theta_t$ is scaled by a factor less than 1 (assuming $\eta\lambda/(\sqrt{v_t} + \epsilon) > 0$), leading to a reduction in $\theta_t$. If $\lambda$ is large, the scaling factor can be significantly less than 1, causing $\theta_t$ to diminish rapidly. This aggressive shrinking overshadows the limited gradient from the loss function, effectively halting meaningful updates aimed at minimizing $L(\theta)$.

**AdamW's Update Mechanism Under Small Loss.** In AdamW, the update rule is:

$$\theta_{t+1} = \theta_t - \eta \frac{\hat{m}_t}{\sqrt{\hat{v}_t} + \epsilon} - \eta \lambda \theta_t$$

Even when $L(\theta_t)$ is small, the gradient-based update term $-\eta \frac{\hat{m}_t}{\sqrt{\hat{v}_t}+\epsilon}$ remains focused on minimizing the loss, while the weight decay term $-\eta \lambda \theta_t$ independently enforces regularization.

This separation ensures that:

- The optimization process remains primarily influenced by the loss gradient.
- Weight decay controls the magnitude of the parameters without dictating the direction of updates.

*Thus, AdamW allows the model to continue optimizing $L(\theta_t)$ effectively, even when the loss is already minimal, while maintaining controlled regularization through weight decay.*

# F  PYTORCH IMPLEMENTATION CODE

```python
# num_class : number of classes
# output : tensor of model outputs
# soft_label : Tensor, shape=[bsz, C]
# hard_label : Tensor, shape=[bsz, 1]
# alpha : smoothing parameter for label smoothing

def GIFT(num_class, output , soft_label, hard_label, alpha):
    # apply label smoothing to hard label
    smooth_label = label_smoothing(hard_label, num_class, alpha)

    # refine soft label
    soft_label = F.normalize(soft_label, dim=1)
    smooth_label = F.normalize(smooth_label, dim=1)
    refined_soft_labels = weight * smooth_label + (1 - weight) * soft_label

    # calculate the coisen similariy
    loss = F.cosine_similarity(output, refined_soft_labels, dim=1)
    return loss
```

Figure 13: Pytorch Implementation Code

