# OpenReview forum: "GIFT: Unlocking Full Potential of Labels in Distilled Dataset at Near-zero Cost"
_ICLR.cc/2025/Conference — ICLR 2025 Poster_

### Official Review · Reviewer_twQ2 · 2024-10-27

**Soundness:** 3
**Presentation:** 3
**Contribution:** 2
**Rating:** 6
**Confidence:** 5

**Summary:**

This paper proposes a simple but effective plug-and-play dataset distillation method to leverage both hard and soft labels at near-zero cost.
It identifies that existing dataset distillation methods are highly sensitive to different loss functions, highlighting the need for a universal loss function. Based on this insight, GIFT introduces a label refinement process that incorporates hard label smoothing, thereby correcting any erroneous signals from the teacher model and enhancing inter-class separation. Additionally, the authors use a cosine similarity-based loss function to fully exploit label information during training.

**Strengths:**

1. Unlike previous methods, this paper shifts focus from the distillation process itself to the model training phase using synthetic data, presenting a novel perspective for dataset distillation. The paper proposes a universal loss function that enhances the performance of most dataset distillation methods.
2. Currently, most state-of-the-art methods incorporate soft labels into distillation, complicating unified evaluation when training models with synthetic data. The approach in this paper could potentially help establish a standardized evaluation benchmark for dataset distillation products.
3. Extensive experimental results demonstrate that GIFT significantly improves state-of-the-art dataset distillation methods on large-scale datasets, such as ImageNet-1K, while also enhancing cross-optimizer and cross-architecture generalization. GIFT’s plug-and-play nature makes it versatile across various datasets and network architectures.
4. The presentation of the paper is great. It provides a detailed discussion and clearly showcases the motivation and methodology.

**Weaknesses:**

1. Although the method offers a novel perspective, its lack of improvement in the distillation process itself somewhat limits the contribution of the paper.
2. I suggest the authors consider incorporating GIFT into [1], i.e., replacing the loss with GIFT's loss in [1], or adding a loss term in other trajectory-matching methods (as long as they can access a well-trained model during training), I am interested to see if this combination could further improve the effectiveness of dataset distillation.


[1] Can pre-trained models assist in dataset distillation? arXiv:2310.03295

**Questions:**

Please refer to weaknesses.

---

> ### Author Response · Authors · 2024-11-20
> **Response to Reviewer twQ2 (1/2)**
>
> > W1: Although the method offers a novel perspective, its lack of improvement in the distillation process itself somewhat limits the contribution of the paper.
> >
>
> We agree with the reviewer that the proposed method does not modify the distillation process itself. However, we would like to emphasize that the primary aim of this work is not to modify the distillation process directly. Instead, our method functions ***as a plug-and-play module*** designed to integrate effortlessly with current dataset distillation techniques, thereby ***enhancing the performance of any existing distilled data.***
>
> Furthermore, it is crucial to highlight that our GIFT module ***incurs no additional computational cost while providing significant performance improvements***. The enhancements achieved through our GIFT module include:
>
> - **Improvements achieved through our GIFT can *surpass* those obtained by modifying the distillation process.**
>     - For instance, on CIFAR-100 with IPC values of 10 and 50, FreD [N1] achieves a marginal improvement of 0.2% and 0.1%, respectively. Similarly, SeqMatch [N2] shows an improvement of 0.3% and 0.5%. Moreover, the method in [N3] improves performance by 1.0% and 1.4%. In contrast, our method demonstrates a significant enhancement of 3.7% and 2.5%, respectively.
>     - Additionally, the aforementioned methods can not distill large-scale datasets such as ImageNet-1k, whereas our method achieves consistent improvements on such datasets.
> - **Strong Robustness to Cross-Architecture and Cross-Optimizer Challenges**: Beyond the state-of-the-art comparisons, our GIFT brings noteworthy improvements in both cross-architecture (Table 7 on page 8) and cross-optimizer generalization (Table 8 on page 8 and Table 16 on page 18). Notably, we reveal that conventional loss functions exhibit significant robustness deficiencies when employed across various optimizers. This critical challenge is often neglected in existing dataset distillation research.
>     - For example, on ImageNet-1K with IPC = 10, GIFT improves the cross-optimizer generalization of the SOTA method RDED by **30.08%** on ResNet-18.
>     - We empirically and theoretically analyze why optimizers exert a significant influence on the performance of various distilled datasets, as elaborated in [Response [Q1] for Reviewer vaeM](https://openreview.net/forum?id=FoF5RaA3ug&noteId=G32Ey8N3PO).
> - **Significant Improvements on Large-scale Dataset:** Our GIFT method can achieve substantial performance gains when applied to complex datasets. For instance, with RDED, our methods can achieve significant improvements (as listed below). These results highlight the method's meaningfulness and potential for further exploration.
>     - On ImageNet-1K with IPC = 10, our method yields 3.9% and 1.8% improvement on ConvNet and ResNet-18, respectively, without incurring additional costs.
> - Beyond state-of-the-art experiments, our ***applications in continual learning*** have shown promising results.
>     - For example, our method enhances the state-of-the-art dataset distillation approach RDED by 1.8%.
>
> Hence, this paper provides valuable advancements in the field of dataset distillation and is highly impactful.  A detailed summary of the contribution and novelty can be found in [Summary of Contribution and Novelty](https://openreview.net/forum?id=FoF5RaA3ug&noteId=z3nUHYAFvP).
>
> [N1] Frequency domain-based dataset distillation. NeurIPS, 2023.
>
> [N2] Sequential subset matching for dataset distillation. NeurIPS, 2023.
>
> [N3] Exploiting Inter-sample and Inter-feature Relations in Dataset Distillation. CVPR, 2024.

---

> ### Author Response · Authors · 2024-11-20
> **Response to Reviewer twQ2 (2/2)**
>
> > W2: I suggest the authors consider incorporating GIFT into [1], i.e., replacing the loss with GIFT's loss in [1], or adding a loss term in other trajectory-matching methods (as long as they can access a well-trained model during training), I am interested to see if this combination could further improve the effectiveness of dataset distillation.
> >
>
> Thank you for your insightful feedback. We have conducted these experiments, and the results clearly demonstrate that ***our GIFT can enhance the effectiveness of the dataset distillation.***
>
> - **Experimental Setup:**
>     - **Dataset and Network:** Specifically, following the experimental setup in [1], we replace the loss term from [1] with our GIFT to generate distilled data via DC on both CIFAR-10 and CIFAR-100 datasets, employing a ConvNet architecture.
>     - **Pre-trained models:** We use various pre-trained models (refer to ‘PTMs’ in [1]) with different initialization parameters and architectures. For instance, experiments with Nm=10 and Na=1 indicate that pre-trained models are initialized with 10 different parameter sets while using the same architecture.  Experiments with Nm = 1 and Na = 4 investigate the influence of model architecture on distilled datasets.
> - **Results:** The results presented in Table X1 illustrate that our GIFT method significantly improves the performance of DC, ***verifying the effectiveness of our GIFT incorporated into the dataset distillation process.***
> - **Why Can Our GIFT Enhance the Distillation Process?**
>     - **GIFT can also improve the effectiveness of hard label utilization:**  We have conducted experiments on both distilled and randomly selected datasets with hard labels to verify whether our GIFT is effective across various data types. The experimental setup and results are detailed in [Response [Q1] for Reviewer izfT](https://openreview.net/forum?id=FoF5RaA3ug&noteId=wRvq8i5jzb).
>     - **GIFT Correct Soft Labels Errors:**  The label refinement mechanism in our GIFT addresses potential inaccuracies in soft labels derived from pre-trained models $\theta^*$ as described in Eq (5) in [1].
>         - *Why might soft labels introduce potential errors?* Despite pre-training on the original dataset, these models exhibit suboptimal performance. For example, the test accuracies of teacher models are only 61.27% on CIFAR-100.  Consequently, the accuracy of soft labels is constrained by the suboptimal nature of the teacher models.
>         - Our GIFT can integrate hard labels to refine the soft labels. The advantages of label refinement in our GIFT are detailed in [Response [W1] for Reviewer izfT](https://openreview.net/forum?id=FoF5RaA3ug&noteId=Qg3JHq3i8X).
>
> Table X1: Performance comparison of DC using various loss terms on CIFAR-10 and CIFAR-100 datasets with IPC = 10. **Bold** denotes the best results.
>
> |  |  | CIFAR-10 | CIFAR-100 |
> | --- | --- | --- | --- |
> | DC |  | 51.4 ± 0.3 | 28.7 ± 0.3  |
> | DC+CLoM | Nm=1, Na=1 | 51.5 ± 0.3 | 29.4 ± 0.4 |
> |  | Nm=10, Na=1 | 52.0 ± 0.3 | 31.8 ± 0.3 |
> |  | Nm=1, Na=4 | 52.2 ± 0.1 | 30.2 ± 0.2  |
> | DC+GIFT | Nm=1, Na=1 | **51.9 ± 0.2** | **31.2 ± 0.3** |
> |  | Nm=10, Na=1 | **52.4 ± 0.3** | **32.1 ± 0.2** |
> |  | Nm=1, Na=4 | **52.6 ± 0.3** | **31.5 ± 0.3** |

---

> ### Author Response · Authors · 2024-11-25
> **A Kind Reminder for Reviewer twQ2**
>
> Dear Reviewer twQ2,
>
> Thank you for your valuable feedback and thorough review of our paper. Your insights have greatly contributed to refining our work. In response to the specific concerns you raised, we have provided detailed explanations to address each concern comprehensively. Below, we summarize your concerns and our key responses:
>
> - **[W1: Lack of improvement in the distillation process itself]:** We would like to clarify that the primary objective of our work is not to directly modify the distillation process. Instead, our approach introduces a ***plug-and-play module*** designed to integrate effortlessly with existing dataset distillation methods, thereby enhancing the performance of distilled datasets. Notably, the improvements achieved through our method, GIFT, can ***surpass those obtained by directly modifying the distillation process***, all while incurring ***no additional cost***.
> - **[W2: Incorporating GIFT into the distillation process of [1]]:** We have conducted experiments to assess the effectiveness of incorporating GIFT within the distillation framework of [1]. The results demonstrate that ***GIFT significantly enhances the effectiveness of the dataset distillation process.*** Additionally, we provide a ***detailed analysis*** elucidating the mechanisms through which GIFT achieves these improvements, shedding light on its efficacy.
>
> We sincerely appreciate the constructive feedback, which has offered meaningful insights to enhance our work. Thank you again for your thoughtful contributions!
>
> If our rebuttal has adequately addressed your concerns, we kindly request that you consider revising your score accordingly. An increased score is critically important to our work at this stage.
>
> We remain open and glad to any additional questions or feedback you may have. Your efforts and detailed review are greatly appreciated, and we value the opportunity to improve our work based on your input. Thank you once again for your time and consideration. We look forward to your further feedback.
>
> Best regards,
>
> Authors of Paper 10413

---

> ### Comment · Reviewer_twQ2 · 2024-11-26
>
> Thanks for the detailed response, most of my concerns are addressed. I suggest authors involve the added experiments into the revision. I will maintain my score and vote for acceptance.

---

> > ### Author Response · Authors · 2024-11-28
> > **Thank You for Your Positive Review**
> >
> > Dear Reviewer twQ2,
> >
> > Thank you very much for your time and effort in reviewing our work. We greatly appreciate your constructive feedback, as well as your support and recommendation for acceptance. We will carefully incorporate the suggested experiments into our revision to further strengthen the manuscript.
> >
> > Once again, thank you for your valuable advice on improving this work.
> >
> > Best regards,
> >
> > Authors of Paper 10413

---

### Official Review · Reviewer_izfT · 2024-10-29

**Soundness:** 3
**Presentation:** 3
**Contribution:** 2
**Rating:** 6
**Confidence:** 4

**Summary:**

This work focuses on the problem of soft label utilization in dataset distillation. Currently, multiple different loss functions are employed to use the soft labels generated by pre-trained models. The choice of loss function can have large influence on the training effects for different distilled datasets. Therefore, the authors propose to refine the generated soft labels, and adopt a unified cosine similarity-based loss function to leverage the soft label information. The experiments suggest that the proposed method uniformly improve the validation performance on multipld benchmarks.

**Strengths:**

1. The motivation is clear and the designed model has correspondence to the motivation.
2. The experiments results show uniformly performance improvement over multiple distilled datasets and multiple benchmarks.
3. It is valuable to look into the validation for dataset distillation.

**Weaknesses:**

1. The authors claim that GIFT can correct erroneous signals from the teacher model, particularly in cases where the teacher assigns an incorrect label of the highest value. However, there is no qualitative or quantitative result supporting it. Since the $\gamma$ parameter is only set as 0.1, it remains unclear how it can correct potential soft label error. Can the authors summarize how much the soft labels change after the refinement?
2. The authors introduce a new evaluation scenario named cross-optimizer generalization. But it is not defined until the end of the introduction section. It is recommended to raise the issue of poor cross-optimizer generalitzation earlier. Does it serve as one of the motivations, or is it simply empirical results after the method is designed?
3. The authors show in Table 6 the cross-architecture generalization results on Tiny-ImageNet. In many cases the performance gain of large models is on par or lower than that on small models. Is the improvement benefited simply from better training effects of cosine similarity or better generalization?
4. Although the proposed method uniformly improves the validation results with soft labels on different distilled datasets, the soft label requires enormous storage space. While it is feasible to use currently available pre-trained models for popular dataset distillation benchmarks, it is not practical for actual use on new application scenarios. Thus, the overall practicality slightly degrades the value of this work for the dataset distillation society.

**Questions:**

1. How would cosine similarity perform if it is applied directly on hard labels / smoothed labels? Will it still outperform the other compared loss functions?
2. The performance improvement is generally higher on small datasets / poor architectures compared with the other end. Can the authors explain the potential reason?

---

> ### Author Response · Authors · 2024-11-20
> **Response to Reviewer izfT (1/2)**
>
> > W1: The authors claim that GIFT can correct erroneous signals from the teacher model, particularly in cases where the teacher assigns an incorrect label of the highest value. However, there is no qualitative or quantitative result supporting it. Since the parameter $\gamma$ is only set as 0.1, it remains unclear how it can correct potential soft label errors. Can the authors summarize how much the soft labels change after the refinement?
> >
>
> We appreciate your valuable inquiry and would like to address your concerns through the following points:
>
> - **Rationale for Integrating Soft and Hard Labels:**
>     - The necessity of incorporating hard labels into the soft labels generated by teacher models arises from the inherent limitations in the performance of these teacher models. Specifically, the test accuracies of teacher models are ***only 61.27%, 49.73%, and 43.6%*** for CIFAR-100, Tiny-ImageNet, and ImageNet-1K, respectively, when trained on commonly used ConvNet architectures in dataset distillation.
>     - Consequently, the accuracy of soft labels is constrained by the suboptimal nature of the teacher models. To mitigate potential inaccuracies in these soft labels, we integrate hard labels to enhance reliability.
>     - We assess the label accuracy both before and after applying the refinement to verify whether the proposed label refinement can correct soft labels.
> - **Comparison of Label Accuracy Before and After Label Refinement:**
>     - Experimental Details: We recorded the label accuracy before and after refinement across each training epoch for three datasets. The parameter $\gamma$, controlling the integration ratio, is set to 0.1.
>     - Results: As depicted in Figure 10 on page 20 of the revised manuscript, refining soft labels results in ***significant performance improvements of 37.1%, 40.95%, and 71.39% for CIFAR-100, Tiny-ImageNet, and ImageNet-1K, respectively***, highlighting the critical role of the proposed label refinement. We have incorporated the analysis and experimental results into the revised manuscript in lines 499-502 on page 10. We appreciate your valuable question once again.
>
> > W2: The authors introduce a new evaluation scenario named cross-optimizer generalization. But it is not defined until the end of the introduction section. It is recommended to raise the issue of poor cross-optimizer generalitzation earlier. Does it serve as one of the motivations, or is it simply empirical results after the method is designed?
> >
>
> Thank you very much for your valuable advice. We have revealed the challenge earlier in the revised manuscript, specifically on lines 45-48 on page 1. Additionally, a formal definition is provided on lines 137-144 on page 3.
>
> Furthermore, the cross-optimizer challenge stems from ***our motivation*** to develop a loss function that demonstrates uniformity and simplicity across various scenarios, including efficacy in cross-optimizer scenarios.  We provide an empirical and theoretical analysis of why optimizers exert a significant influence on the performance of various distilled datasets, as elaborated in [Response [Q1] for Reviewer vaeM](https://openreview.net/forum?id=FoF5RaA3ug&noteId=G32Ey8N3PO), highlighting the necessity of a uniform loss function.
>
> > W3: The authors show in Table 6 the cross-architecture generalization results on Tiny-ImageNet. In many cases the performance gain of large models is on par or lower than that on small models. Is the improvement benefited simply from better training effects of cosine similarity or better generalization?
> >
>
> Thank you for your valuable question. We kindly suspect there may have been a misunderstanding regarding the table. For the cross-architecture generalization experiments, distilled data is generated using ConvNet and ResNet-18 (denoted as D) and is subsequently evaluated on both small-scale and large-scale networks (denoted as E). *The evaluation results are presented from left to right in the table,* corresponding to Table 6 in the original paper and Table 7 in the revised version.
>
> For large-scale networks, as illustrated on the right side of the table, ***the performance improvements are more pronounced***. For example, when distilled data is generalized on ConvNet and evaluated on the large-scale network Swin-V2-Tiny, the improvement is quantified at 12.2%.
>
> If we have misunderstood your query, please let us know. We sincerely hope to discuss this further with you.

---

> ### Author Response · Authors · 2024-11-20
> **Response to Reviewer izfT (2/2)**
>
> > W4: Although the proposed method uniformly improves the validation results with soft labels on different distilled datasets, the soft label requires enormous storage space. While it is feasible to use currently available pre-trained models for popular dataset distillation benchmarks, it is not practical for actual use on new application scenarios. Thus, the overall practicality slightly degrades the value of this work for the dataset distillation society.
> >
>
> Thank you for your valuable question. We would like to address your points as follows:
>
> - **Prevalence of Soft Labels:** The majority of state-of-the-art data distillation methodologies [1, 2, 3, 4, 5] utilize soft labels, particularly when applied to large-scale datasets. Moreover, [1] highlights that soft labels can enhance training stability. Consequently, *soft labels are necessary for current dataset distillation applications*, and our method offers a valuable exploration into optimizing their use.
> - **Our GIFT can Enhance the Utilization of Hard Labels:** Given that soft labels demand substantial storage space, we conduct experiments to evaluate the effectiveness of our method for hard labels. The experimental setup and results are detailed in [Q1]. These results demonstrate that *our method can also significantly improve the utilization of hard labels.*
>
> [1] Towards lossless dataset distillation via difficulty-aligned trajectory matching. ICLR, 2024.
>
> [2] On the diversity and realism of distilled dataset: An efficient dataset distillation paradigm. CVPR, 2024.
>
> [3] Generalized large-scale data condensation via various backbone and statistical matching. CVPR, 2024.
>
> [4] Squeeze, recover and relabel: Dataset condensation at imagenet scale from a new perspective. NeurIPS, 2023.
>
> [5] Scaling up dataset distillation to imagenet-1k with constant memory. ICML, 2023.
>
>
> > Q1: How would cosine similarity perform if it is applied directly on hard labels / smoothed labels? Will it still outperform the other compared loss functions?
> >
>
> Thank you for your constructive feedback. *Yes, applying our method to both hard labels and smoothed labels results in significant improvements. The detailed experimental setup and results are outlined below:*
>
> **Experimental setting:**
>
> - **Dataset:** To evaluate the efficacy of GIFT across various data types, we conduct experiments on both distilled and randomly selected datasets, employing hard and smoothed labels, with IPC=10.
>     - *Distilled Dataset:* we use the state-of-the-art dataset distillation method, RDED, which uses soft labels generated by teacher models. In the experiments, we maintain the distilled images and replace soft labels with hard and smoothed labels for model training.
>     - *Random Dataset:* we randomly select 10 images for each class from the original dataset. Here, we also employed hard and smoothed labels for training.
> - **Loss Function Comparison:**
>     - For hard labels, we compare the traditional cross-entropy loss (CE) with our method.
>     - For smoothed labels, we compare SoftCE, KL, and MSE loss against our method.
>
> The results for two types of data are presented in Table X1 and Table X2, respectively. It is evident that ***applying our method to label utilization consistently improves performance*** for both the distilled and the randomly selected data, thereby confirming the method's effectiveness.
>
> Thank you for your constructive question again. We have added the valuable experiments to the revised manuscript in lines 513-516 on page 10.
>
> Table X1: Evaluation of Loss Functions Across Hard and Smoothed Labels on the Distilled Data Generated via RDED with IPC=10.
>
> | Label Type | Loss Function | CIFAR100 | Tiny-ImageNet | ImageNet-1K |
> | --- | --- | --- | --- | --- |
> | Hard | CE | 21.6 ± 0.2 | 13.5 ± 0.3 | 8.3 ± 0.4 |
> |  | Ours | **22.8 ± 0.2 (↑1.2)** | 14.8 ± 0.3 (**↑1.3)** | **9.8 ± 0.2 (↑1.5)** |
> | Smoothed | SoftCE | 21.9 ± 0.2 | 13.8 ± 0.2 | 8.5 ± 0.2 |
> |  | KL | 21.7 ± 0.3 | 13.3 ± 0.3 | 8.0 ± 0.3 |
> |  | MSE | 22.1 ± 0.1 | 14.1 ± 0.1 | 8.8 ± 0.3 |
> |  | Ours | **23.1 ± 0.2 (↑1.0)** | **15.2 ± 0.3 (↑1.1)** | **10.2 ± 0.2 (↑1.4)** |
>
>
> Table X2: Evaluation of Loss Functions Across Hard and Smoothed Labels on the Randomly Selected Data with IPC=10.
> | Label Type | Loss Function | CIFAR100 | Tiny-ImageNet | ImageNet-1K |
> | --- | --- | --- | --- | --- |
> | Hard | CE | 18.9 ± 0.3 | 9.4 ± 0.1 | 4.0 ± 0.3 |
> |  | Ours | **20.3 ± 0.1 (↑1.4)** | **11.6 ± 0.1 (↑2.2)** | **5.1 ± 0.2 (↑1.1)** |
> | Smoothed | SoftCE | 19.0 ± 0.3 | 9.8 ± 0.3 | 4.5 ± 0.2 |
> |  | KL | 17.9 ± 0.2 | 8.5 ± 0.3 | 3.9 ± 0.1 |
> |  | MSE | 19.2 ± 0.1 | 9.7 ± 0.2 | 4.6 ± 0.3 |
> |  | Ours | **20.4 ± 0.2 (↑1.2)** | **11.8 ± 0.2 (↑2.0)** | **5.6 ± 0.2 (↑1.0)** |
>
>
> > Q2: The performance improvement is generally higher on small datasets / poor architectures compared with the other end. Can the authors explain the potential reason?
> >
>
> Please refer to Response to [W3].

---

> ### Author Response · Authors · 2024-11-25
> **A Kind Reminder for Reviewer izfT**
>
> Dear Reviewer izfT,
>
> Thank you for your valuable feedback and thorough review of our paper. Your insights have greatly contributed to refining our work. In response to the specific concerns you raised, we have provided detailed explanations to address each concern comprehensively. Below, we summarize your concerns and our key responses:
>
> - **[W1: Necessity of soft label refinements]:** We first explain the reason for integrating hard labels into the refinement of soft labels. Additionally, we compare label accuracy before and after refinement, demonstrating ***a significant improvement post-refinement***. The results highlight the pivotal role of the proposed label refinement.
> - **[W2: Definition of cross-optimizer generalization]:** We address this concern by revealing cross-optimizer generalization earlier and providing a formal definition in the revised manuscript. Furthermore, we provide ***both empirical and theoretical analyses*** to elucidate the substantial impact of optimizers on the performance of distilled datasets.
> - **[W3 & Q2: Performance of cross-architecture generalization]:** We suspect there may have been some misunderstanding regarding the table (Table 7 in the revised paper). For large-scale networks, as shown on the right side of the table, ***the observed performance improvements are notably pronounced***. If we have misunderstood your query, please let us know. We sincerely hope to discuss this further with you.
> - **[W4 & Q1: Applicability of our method to hard and smoothed labels]:** We have conducted experiments on both distilled and randomly selected datasets, applying our approach to both hard and smoothed labels. The results consistently show ***significant performance improvements for both hard and smoothed labels***.
>
> We have incorporated your valuable suggestions into the revised manuscript. Thank you once again for your insightful feedback!
>
> If our rebuttal has adequately addressed your concerns, we kindly request that you consider revising your score accordingly. An increased score is critically important to our work at this stage.
>
> We remain open and glad to any additional questions or feedback you may have. Your efforts and detailed review are greatly appreciated, and we value the opportunity to improve our work based on your input. Thank you once again for your time and consideration. We look forward to your further feedback.
>
> Best regards,
>
> Authors of Paper 10413

---

> > ### Comment · Reviewer_izfT · 2024-11-25
> >
> > Thanks for the reply. The authors have provided extended experiments to address my concerns. Since the method can also improve the training performance with hard label, I believe it is generally helpful to the dataset distillation society. And I have raised the score.

---

> > > ### Author Response · Authors · 2024-11-28
> > > **Thank You for Your Positive Review**
> > >
> > > Dear Reviewer izfT,
> > >
> > > Thank you very much for your effort and time in reviewing our work. We are delighted to hear that our rebuttal effectively addressed your comments and that you have increased your rating. We respectfully believe that this work aligns well with the ICLR community, and we sincerely hope it can be seen by more researchers in the field.
> > >
> > > Once again, we deeply appreciate your thoughtful and constructive feedback!
> > >
> > > Best regards,
> > >
> > > Authors of Paper 10413

---

### Official Review · Reviewer_FuA8 · 2024-10-29

**Soundness:** 3
**Presentation:** 4
**Contribution:** 3
**Rating:** 8
**Confidence:** 4

**Summary:**

This paper introduces a novel method for dataset distillation that fully leverages soft labels generated by pre-trained teacher models. The authors highlight the critical role of selecting the right loss function, showing that the model's performance on synthetic datasets is highly dependent on this choice. They propose **GIFT**, a simple yet effective method that refines soft labels and uses a cosine similarity-based loss function to boost performance across various tasks. The approach also enhances generalization across optimizers and architectures. Extensive experiments demonstrate that **GIFT** significantly improves performance without adding computational costs, making it scalable even for large datasets like ImageNet-1K.

**Strengths:**

[S1] The paper dives into a crucial part of dataset distillation, specifically how to choose effective loss functions for label utilization in DD frameworks.
[S2] The writing is clear and easy to follow.
[S3] The use of cosine similarity is backed by solid theoretical reasoning.
[S4] The ablation studies are detailed, and the method shows promising results in continual learning.
[S5] GIFT consistently improves state-of-the-art dataset distillation methods across various datasets without adding any extra computational cost.

**Weaknesses:**

[W1] From a technical standpoint, the contribution of the paper feels somewhat limited. Both the hard-label refinement approach and the cosine-similarity loss function have already been explored in many knowledge distillation studies. Additionally, the paper lacks convincing evidence or justification that these techniques are specifically suited for the synthetic datasets used in dataset distillation.

[W2] Regarding the experimental results, the improvements brought by the proposed label utilization algorithm to existing DD methods are relatively modest. However, since the approach incurs no additional computational cost and offers general performance boosts, the results are still reasonable and acceptable.

**Questions:**

- Could you share the results of your label utilization method on a randomly selected data subset?
- How does GIFT perform with higher IPC values, such as 100 or 200?
- Is it possible to achieve lossless dataset distillation using GIFT on any dataset?
- It would be interesting to see how the Jensen-Shannon divergence loss function performs within the GIFT framework.

---

> ### Author Response · Authors · 2024-11-20
> **Response to Reviewer FuA8 (1/3)**
>
> > W1: From a technical standpoint, the contribution of the paper feels somewhat limited. Both the hard-label refinement approach and the cosine-similarity loss function have already been explored in many knowledge distillation studies. Additionally, the paper lacks convincing evidence or justification that these techniques are specifically suited for the synthetic datasets used in dataset distillation.
> >
>
> We agree with the reviewer that label refinement and cosine-similarity loss have been discussed in some literature on other domains. Nonetheless, to the best of our knowledge, our paper is the first to ***provide a novel perspective and solid theory*** of label utilization in distilled data, as highlighted and acknowledged by Reviewers vaeM and twQ2. Importantly, ***our proposed method does not incur additional costs***. A detailed summary of our contributions and the novelty of this paper are provided in [Summary of Contribution and Novelty](https://openreview.net/forum?id=FoF5RaA3ug&noteId=z3nUHYAFvP).
>
>
> Regarding your concerns about why these techniques are specifically suited for distilled datasets, we would like to address them by analyzing the inherent properties of distilled data:
>
> - **Highly Informative Nature of Distilled Dataset:** Recent studies [1, 2, 3] have demonstrated that distilled datasets are highly informative due to the soft labels that are assigned by pre-trained models, resulting in high-dimensional vectors that are approximately orthogonal to each other [4, 5, 6, 7]. Therefore, to minimize the upper bound of training loss as shown in Equation (2) on page 4, ***derived from theoretical considerations***, we propose a loss function Equation (3) (cosine-similarity loss). A detailed explanation is provided in lines 209-221 on page 4.
> - **Extremely Small Size of Distilled Dataset:** Another fundamental characteristic of the distilled dataset is its typically small size, with IPC = {1, 10, 50}. To demonstrate that the specific cosine-similarity loss is particularly effective for small distilled datasets, *we compare our method with various loss functions.*
>     - We compare it with the *knowledge distillation algorithm*, as shown in Table 5 on page 7, with analysis provided in lines 353-357 on page 7. These loss functions are shown to be unsuitable for small distilled datasets.
>     - Furthermore, we compare our approach with *commonly used loss functions*, such as MSE, as illustrated in Table 9 on page 9. The results indicate that our method consistently outperforms others.
>     - Therefore, extensive experiments verify that the cosine-similarity loss is particularly effective for distilled datasets.
>
> Based on both theoretical analysis and comprehensive empirical studies, we can conclude that our GIFT is particularly effective for distilled datasets.
>
> [1] Efficiency for Free: Ideal Data Are Transportable Representations, NeurIPS 2025.
>
> [2] Squeeze, recover and relabel: Dataset condensation at imagenet scale from a new perspective. NeurIPS, 2023.
>
> [3] On the diversity and realism of distilled dataset: An efficient dataset distillation paradigm. CVPR, 2024.
>
> [4] Learning Neural Networks with Sparse Activations. COLT, 2024.
>
> [5] White-box transformers via sparse rate reduction. NeurIPS, 2023.
>
> [6] On the principles of parsimony and self-consistency for the emergence of intelligence. Frontiers of Information Technology & Electronic Engineering, 2022.
>
> [7] High-dimensional statistics: A non-asymptotic viewpoint. Cambridge university press, 2019.

---

> ### Author Response · Authors · 2024-11-20
> **Response to Reviewer FuA8 (2/3)**
>
> > W2: Regarding the experimental results, the improvements brought by the proposed label utilization algorithm to existing DD methods are relatively modest. However, since the approach incurs no additional computational cost and offers general performance boosts, the results are still reasonable and acceptable.
> >
>
> We appreciate your acknowledgment of our contributions and wish to emphasize that our proposed method demonstrates substantial enhancements in several challenging scenarios.
>
> - For the ***cross-architecture and cross-optimizer challenges,***  GIFT significantly enhances the state-of-the-art methods.
>     - For example, on ImageNet-1K with IPC = 10, GIFT improves the cross-optimizer generalization of the SOTA method RDED by **30.08%** on ResNet-18.
>     - Reviewer twQ2 specifically emphasized the significance of the improvements in cross-optimizer generalization.
>     - We empirically and theoretically analyze why optimizers exert a significant influence on the performance of various distilled datasets, as elaborated in [Response [Q1] for Reviewer vaeM](https://openreview.net/forum?id=FoF5RaA3ug&noteId=G32Ey8N3PO).
> - **Significant Improvements on Large-scale Dataset:** Our GIFT method can achieve substantial performance gains when applied to complex datasets. For instance, with RDED, our methods can achieve significant improvements (as listed below). These results highlight the method's meaningfulness and potential for further exploration.
>     - When IPC=10, our method yields 3.9% and 1.8% improvement on ConvNet and ResNet-18, respectively, without incurring additional costs.
> - Beyond state-of-the-art experiments, our ***applications in continual learning*** have shown promising results.
>     - For example, our method enhances the state-of-the-art dataset distillation approach RDED by 1.8%.
>
> Notably, these results can be achieved without any additional cost, underscoring the significance of this paper.
>
>
>
> > Q1: Could you share the results of your label utilization method on a randomly selected data subset?
> >
>
> Thanks for your valuable advice. Following the suggestion, we have conducted a series of experiments that randomly select subset data when IPC=10, employing ***three types of labels*** across datasets of varying scales and resolutions.
>
> - **Loss Function Comparison:**
>     - For hard labels: we compare the traditional cross-entropy loss (CE) with our method.
>     - For soft labels: we compare with SoftCE, KL, and MSE loss.
>     - For smoothed labels: they are derived from the original hard labels. We compare with SoftCE, KL, and MSE loss.
>
> The results are presented in Table X1. It is evident that *applying our method to label utilization consistently improves performance* for both the hard and soft labels of randomly selected data.
>
> Table X1: Evaluation of Loss Functions Across Three Label Types on a Randomly Selected Subset with IPC=10.
>
> | Label Type | Loss Function | CIFAR100 | Tiny-ImageNet | ImageNet-1K |
> | --- | --- | --- | --- | --- |
> | Hard | CE | 18.9 ± 0.3 | 9.4 ± 0.1 | 4.0 ± 0.3 |
> |  | Ours | **20.3 ± 0.1 (↑1.4)** | **11.6 ± 0.1 (↑2.2)** | **5.1 ± 0.2 (↑1.1)** |
> | Soft | SoftCE | 41.5 ± 0.3 | 35.7 ± 0.3 | 13.3 ± 0.1 |
> |  | KL | 45.4 ± 0.3 | 38.1 ± 0.2 | 15.6 ± 0.2 |
> |  | MSE | 46.5 ± 0.2 | 38.5 ± 0.1 | 15.8 ± 0.2 |
> |  | Ours | **49.3 ± 0.1 (↑2.8)** | **40.3 ± 0.3 (↑1.8)** | **16.6 ± 0.2 (↑0.8)** |
> | Smoothed | SoftCE | 19.0 ± 0.3 | 9.8 ± 0.3 | 4.5 ± 0.2 |
> |  | KL | 17.9 ± 0.2 | 8.5 ± 0.3 | 3.9 ± 0.1 |
> |  | MSE | 19.2 ± 0.1 | 9.7 ± 0.2 | 4.6 ± 0.3 |
> |  | Ours | **20.4 ± 0.2 (↑1.2)** | **11.8 ± 0.2 (↑2.0)** | **5.6 ± 0.2 (↑1.0)** |
>
>
>
> > Q2: How does GIFT perform with higher IPC values, such as 100 or 200?
> >
>
> Thank you for your question. We assume that the reviewer might miss the experimental results with higher IPC values. Indeed, we have conducted some experiments with higher IPC values.
>
> - **Comparison with State-of-the-Art Dataset Distillation Methods:**
>     - We evaluate our method on Tiny-ImageNet with IPC=100 and ImageNet-1K with IPC values of 100 and 200.
>     - These results are detailed in Table 3 on page 6, and the corresponding analysis is presented in lines 323-341 on page 6. *Our method consistently yields substantial performance improvements*. For example, on Tiny-ImageNet with IPC=100, our method can achieve a performance gain of 3.1%.
> - **Comparison with Other Loss Functions:**
>     - Moreover, we conduct experiments on ImageNet-1K with IPC=100 to compare with knowledge distillation loss functions, detailed in Table 5 on page 7, and with commonly used loss functions in dataset distillation, shown in Table 9 on page 9.
>     - These experimental results consistently indicate that *our method provides substantial performance improvements.*
>
> In summary, these results clearly indicate that ***our method provides notable performance improvements at higher IPC values.***

---

> ### Author Response · Authors · 2024-11-20
> **Response to Reviewer FuA8 (3/3)**
>
> > Q3: Is it possible to achieve lossless dataset distillation using GIFT on any dataset?
> >
>
> Yes, we have conducted experiments on varying scales and resolutions, from small-scale, low-resolution datasets (e.g., CIFAR-100) to large-scale, high-resolution datasets (e.g., ImageNet-1K) on various architectures.
>
> Specifically, we evaluate the performance of distilled datasets with IPC values set at 50 for CIFAR-10, 100 for Tiny-ImageNet, and 200 for ImageNet-1K, compared to their full original datasets. As illustrated in Table X2, the results clearly indicate that the performance gap between the distilled datasets, evaluated via our GIFT approach, and the full original datasets is ***negligible***. For instance, the performance gap between the distilled ImageNet-1K and the full ImageNet-1K dataset is only 0.9%. Consequently, the results substantiate that our GIFT approach achieves near-lossless dataset distillation.
>
> Table X2: Performance comparison between the full dataset and the distilled dataset.
> |  | CIFAR-100 | Tiny-ImageNet | ImageNet-1K |
> | --- | --- | --- | --- |
> | Full Dataset | 66.6 ± 0.3 | 61.8 ± 0.2 | 64.8 ± 0.2 |
> | Distilled Dataset | 64.4 ± 0.1  | 59.2 ± 0.3 | 63.4 ± 0.2 |
> | Distilled Dataset (w/ GIFT) | 65.3 ± 0.2 | 60.6 ± 0.2 | 63.9 ± 0.1 |
>
>
> > Q4: It would be interesting to see how the Jensen-Shannon divergence loss function performs within the GIFT framework.
> >
>
> Thank you for your constructive advice. We have conducted additional experiments to evaluate our approach with the Jensen-Shannon divergence loss function (JS), as presented in Table X3. Moreover, we have incorporated the results into the revised manuscript in lines 472-474 on page 9, highlighted in red text. We appreciate your valuable question once again.
>
> - **Comparison Methods:** Specifically, our GIFT framework incorporates a label refinement strategy and a cosine similarity loss. Therefore, we conduct comparisons with the Kullback-Leibler (KL) divergence loss, JS loss, JS loss combined with label refinement, and our GIFT.
> - **Results:** As reported in the table, ***our GIFT model consistently demonstrates superior performance.*** Notably, JS loss can yield marginal improvements. Furthermore, when it is combined with label refinement, additional enhancements are achieved, highlighting the efficacy of our proposed label refinement.
>
> Table X3: Comparison with JS loss. JS means Jensen-Shannon divergence loss function. The experiment is conducted on distilled datasets generated via RDED [1].
> |  | CIFAR100 |  | Tiny-ImageNet |  | ImageNet-1K |  |  |
> | --- | --- | --- | --- | --- | --- | --- | --- |
> |  | 10 | 50 | 10 | 50 | 10 | 50 | 100 |
> | KL | 47.5 ± 0.3 | 55.7 ± 0.4 | 41.4 ± 0.3 | 47.2 ± 0.1 | 20.1 ± 0.4 | 38.5 ± 0.4 | 41.8 ± 0.2 |
> | JS | 47.9 ± 0.1 (↑ 0.4) | 55.9 ± 0.3 (↑ 0.2) | 41.8 ± 0.2 (↑ 0.4) | 47.3 ± 0.2 (↑ 0.1) | 20.5 ± 0.3 (↑ 0.4) | 38.6 ± 0.3 (↑ 0.1) | 41.9 ± 0.3 (↑ 0.1) |
> | JS + Label Refine | 48.2 ± 0.3 (↑ 0.7) | 56.3 ± 0.3 (↑ 0.6) | 42.1 ± 0.2 (↑ 0.7) | 47.3 ± 0.1 (↑ 0.1) | 21.3 ± 0.2 (↑ 1.2) | 38.9 ± 0.3 (↑ 0.4) | 42.2 ± 0.2 (↑ 0.4) |
> | **GIFT (ours)** | **50.6 ± 0.3 (↑ 3.1)** | **57.9 ± 0.2 (↑ 2.2)** | **44.0 ± 0.2 (↑ 2.6)** | **48.3 ± 0.1 (↑ 1.1)** | **24.0 ± 0.8 (↑ 3.9)** | **39.5 ± 0.1 (↑ 1.0)** | **42.5 ± 0.3 (↑ 0.7）** |
>
> [1] On the diversity and realism of distilled dataset: An efficient dataset distillation paradigm. CVPR, 2024.

---

> > ### Comment · Reviewer_FuA8 · 2024-11-22
> >
> > First of all, I appreciate the authors' response. They addressed most of my comments, except for the lossless issue, although the results are close to being lossless. Overall, the paper is strong, and based on the revisions, I have decided to increase my rating to accept.

---

> > > ### Author Response · Authors · 2024-11-23
> > > **Thank You for Your Positive Review**
> > >
> > > Dear Reviewer FuA8,
> > >
> > > Thank you for your timely response! We are delighted to hear that our rebuttal effectively addressed your comments and that you have increased your rating to accept (8). We respectfully believe that this work aligns well with the ICLR community, and we sincerely hope it can be seen by more researchers in the field.
> > >
> > > Once again, we deeply appreciate your thoughtful and constructive feedback!
> > >
> > > Best regards,
> > >
> > > Authors of Paper 10413

---

### Official Review · Reviewer_vaeM · 2024-11-02

**Soundness:** 3
**Presentation:** 3
**Contribution:** 3
**Rating:** 6
**Confidence:** 4

**Summary:**

This paper conducts a comprehensive comparison of various loss functions for soft label utilization in DD and find that the model trained on the synthetic dataset exhibits high sensitivity to the choice of loss function for soft label utilization. Building on these insights, the authors introduce GIFT, which encompasses soft label refinement and a cosine similarity-based loss function to use full label information. Extensive experiments demonstrate the effectiveness of GIFT.

**Strengths:**

- It is interesting to find that models trained on synthetic datasets show high sensitivity to the choice of loss function for soft label utilization.
- The authors reveal that traditional loss functions suffer from significant robustness deficiencies when applied across different optimizers.
- They propose GIFT and prove its effectiveness through experiments.
- The experiment is well designed.

**Weaknesses:**

- In Table 1, the results of ResNet-18 on CIFAR-100 are not very significant, which may not reflect the effect of the proposed algorithm.
- The current synthetic datasets on Tiny-ImageNet and ImageNet-1K are only generated using Conv-4. Can we experiment with structures such as transformer?

**Questions:**

- Can you explain theoretically why different optimizers have such a big impact on DD?

---

> ### Author Response · Authors · 2024-11-20
> **Response to Reviewer vaeM (1/2)**
>
> > W1: In Table 1, the results of ResNet-18 on CIFAR-100 are not very significant, which may not reflect the effect of the proposed algorithm.
> >
>
> Thank you for your insightful feedback. We acknowledge that the improvements for ResNet-18 on CIFAR-100 are less pronounced than those for ConvNet. We would like to mitigate your concerns from the following points:
>
> - **Primary Reason:** when trained on distilled data, larger models like ResNet-18 ***already*** achieve strong performance, inherently limiting the observable enhancements from our proposed algorithm.
>     - For example, with IPC=50, the distilled dataset generated via RDED achieves an overall performance of 64.4% using ResNet-18, compared to 66.6% when using the entire original dataset. Our method attains a performance of 65.3%, representing a 0.9% improvement. Given these results, *it is reasonable to expect diminishing returns from stronger models.*
> - **Broader Impact and Significance of our GIFT:**
>     - **No Cost of our GIFT**: GIFT enhances performance without incurring any extra computational overhead, making it an efficient solution.
>     - **Uniformity and Simplicity**: As a plug-and-play module, GIFT integrates with existing systems to improve label utilization universally across various datasets, as validated by extensive experiments and Reviewer twQ2.
>     - **Strong Robustness to Cross-Architecture and Cross-Optimizer Challenges**: Beyond the state-of-the-art comparisons, our GIFT brings noteworthy improvements in both cross-architecture (Table 7 on page 8) and cross-optimizer generalization (Table 8 on page 8 and Table 16 on page 18), the latter of which is often overlooked in current dataset distillation research.
>         - For example, on ImageNet-1K with IPC = 10, GIFT improves the cross-optimizer generalization of the SOTA method RDED by **30.08%** on ResNet-18.
>         - Reviewer twQ2 specifically emphasized the significance of the improvements in cross-optimizer generalization.
>         - We ***empirically and theoretically*** analyze why optimizers exert a significant influence on the performance of various distilled datasets.
>     - **Significant Improvements on Large-scale Dataset:** Our GIFT method can achieve substantial performance gains when applied to complex datasets. For instance, with RDED, our methods can achieve significant improvements (as listed below). These results highlight the method's meaningfulness and potential for further exploration.
>         - ConvNet: 3.9% improvement for IPC=10 and 1.0% improvement for IPC=50
>         - ResNet-18: 1.8% improvement for IPC=10 and 1.0% improvement for IPC=50
>
> Notably, these results can be achieved ***without any additional cost***, underscoring the significance of this paper.
>
>
> > W2: The current synthetic datasets on Tiny-ImageNet and ImageNet-1K are only generated using Conv-4. Can we experiment with structures such as transformer?
> >
>
> Thank you for your valuable question. We would like to address your points as follows:
>
> 1. **ResNet-18 Was Used in (Tiny-)ImageNet Experiments:** We would like to clarify that the distilled data used in our experiments is also generated using ResNet-18, not solely Conv-4, such as the state-of-the-art comparison experiments presented in Tables 1, 2, and 3 on page 6. Specifically, in Table 1, when the network is indicated as the network, it signifies that the distilled data is both generated and evaluated on ResNet-18. We have provided a more detailed description of the distilled data in the revised manuscript to improve the clarity of our paper, as detailed in lines 258-259 on page 5.
> 2. **Limitations of Current Data Distillation Methods (Not Our Method):** Current data distillation methods primarily employ Conv-4 and ResNet-18 architectures, with no implementations involving transformers. Our approach is a plug-and-play module that leverages existing datasets without engaging in a distillation process, thus preventing us from conducting experiments with transformer-generated distilled data.
> 3. **Additional Experiments with Transformers:** We follow the reviewer’s suggestion and conduct additional experiments using distilled data generated by the state-of-the-art dataset distillation method RDED for Tiny-ImageNet and ImageNet-1K with Swin Transformer architectures.
>     - The results, presented in Table X1, demonstrate significant performance improvements, highlighting the efficacy and potential of our method. We have added the valuable experiments to the revised manuscript in lines 342-345 on page 7, highlighted in red text.
>
> Table X1: Top-1 accuracy (%) on Tiny-ImageNet and ImageNet-1K with IPC={10, 50} Using Swin Transformer.
> |  |Tiny-ImageNet (IPC=10) | Tiny-ImageNet (IPC=50) | ImageNet-1K (IPC=10) | ImageNet-1K (IPC=50) |
> | --- | --- | --- | --- | --- |
> | RDED | 47.1 ± 0.3 | 55.1 ± 0.3 | 42.3 ± 0.2 | 58.6± 0.1 |
> | RDED+Ours | **48.6± 0.2 (↑1.5)** | **56.2± 0.2 (↑1.1)** | **43.5 ± 0.2 (↑1.2)** | **59.4± 0.2 (↑0.8)** |

---

> ### Author Response · Authors · 2024-11-20
> **Response to Reviewer vaeM (2/2)**
>
> > Q1: Can you explain theoretically why different optimizers have such a big impact on DD?
> >
>
> Thank you for your insightful question. We will explain the substantial impact of different optimizers on distilled data evaluation ***from both empirical and theoretical perspectives***. Additionally, we have incorporated these analyses into the revised manuscript in Appendix E on pages 21-25. Thank you again for your valuable advice.
>
> We will begin by analyzing the three optimizers, namely:
>
> - **SGD** directly computes gradients from loss values, significantly impacting the update of model parameters.
> - **Adam** includes weight decay in the gradient computation, meaning that *when the primary gradient signal (from the loss) is small, weight decay can overshadow it, leading to ineffective updates toward minimizing the loss.*
> - **AdamW** separates the concerns of optimization and regularization. By applying weight decay independently, it ensures that the optimization process remains focused on minimizing the loss function, while regularization acts as a controlled adjustment to the parameter magnitudes.
>
> Optimizers inherently exhibit diverse characteristics, resulting in distinct training dynamics when applied to distilled datasets generated by various methods. Specifically, *these distilled datasets, trained with varying loss functions, **exhibit distinct loss values.*** We reveal that the performance of the optimizers is ***highly influenced*** by these loss values, as demonstrated by the subsequent empirical evidence and theoretical analysis.
>
> - **Empirical Perspective:**
>     - **Preliminary Finding:** When employing the KL divergence loss function, the RDED generally exhibits low loss values, as depicted in Figure 11 on page 21 of the revised manuscript. Notably, our GIFT framework does not achieve small loss values.
>     - **Experimental Setup:** To examine the impact of loss values on optimizer performance, we conduct experiments utilizing RDED-generated distilled data. We train models using three distinct optimizers, each with distinct hyperparameter configurations, and compute the gradient norm for each. Specifically, we varied the learning rate for the SGD optimizer and modified the weight_decay parameter for both the Adam and AdamW optimizers.
>     - **Results:** The gradient norms of these models, presented in Figure 12 on page 21 of the revised manuscript, demonstrate that ***when loss values are small, the training dynamics exhibit heightened sensitivity to the optimizer choice.***
>     - **Conclusion:** Therefore, the performances of different optimizers are highly influenced by the loss values. Notably, our GIFT framework can not obtain small loss values, achieving robust performance across various optimizers.
> - **Theoretical Analysis:** We provide a comprehensive theoretical analysis in Appendix E.2 on pages 22-25 of the revised manuscript.

---

> ### Author Response · Authors · 2024-11-25
> **A Kind Reminder for Reviewer vaeM**
>
> Dear Reviewer vaeM,
>
> Thank you for your insightful feedback and thorough review of our paper. Your comments have been instrumental in enhancing our work. In response to the specific concerns you raised, we have provided detailed explanations to address each concern comprehensively. Below, we summarize your concerns and our key responses:
>
> - **[W1: The results of ResNet-18 on CIFAR-100 are not very significant]:** We have thoroughly ***analyzed*** this observation and would like to emphasize that our GIFT demonstrates ***significant improvements*** across key scenarios, including large-scale datasets, cross-architecture evaluations, and cross-optimizer challenges. Importantly, these gains are achieved ***without*** ***incurring any additional cost.***
> - **[W2: Experiment with architectures such as transformers]:** We have conducted experiments using transformer architectures. The results demonstrate that ***our method achieves consistent and significant improvements***.
> - **[Q1: Theoretical analysis regarding cross-optimizer challenges]:** We have analyzed the substantial impact of different optimizers on distilled data evaluation ***from both empirical and theoretical perspectives***.
>
> We have incorporated your valuable suggestions into the revised manuscript. Thank you once again for your insightful feedback!
>
> If our rebuttal has adequately addressed your concerns, we kindly request that you consider revising your score accordingly. An increased score is critically important to our work at this stage.
>
> We remain open and glad to any additional questions or feedback you may have. Your efforts and detailed review are greatly appreciated, and we value the opportunity to improve our work based on your input. Thank you once again for your time and consideration. We look forward to your further feedback.
>
> Best regards,
>
> Authors of Paper 10413

---

> > ### Comment · Reviewer_vaeM · 2024-11-27
> >
> > Thanks for your detailed response, most of my concerns are addressed. I suggest authors involve the added experiments into the revision. I will maintain my score and vote for acceptance.

---

> > > ### Author Response · Authors · 2024-11-28
> > > **Thank You for Your Positive Review**
> > >
> > > Dear Reviewer vaeM,
> > >
> > > Thank you very much for your effort and time in reviewing our work. We greatly appreciate your constructive feedback, as well as your support and recommendation for acceptance. We will carefully incorporate the suggested experiments into our revision to further strengthen the manuscript.
> > >
> > > Once again, thank you for your valuable advice and contribution to improving our work.
> > >
> > > Best regards,
> > >
> > > Authors of Paper 10413

---

### Author Response · Authors · 2024-11-20
**General Response**

We sincerely thank the reviewers for their insightful feedback. We are delighted that the reviewers acknowledged that our ***presentation*** is excellent and easy to follow (Reviewers FuA8, twQ2), ***motivation*** is clear and novel (Reviewers FuA8, twQ2,izfT), ***findings*** are crucial and interesting (Reviewers vaeM and FuA8), the newly ***revealed challenge*** is important (Reviewer vaeM), ***method*** has solid theoretical foundation without any extra cost (Reviewer FuA8), ***experiments*** are extensive and well-designed (Reviewers vaeM and twQ2), the ***improvements*** are significant and consistent (All Reviewers vaeM, FuA8, izfT, twQ2).

Furthermore, our contributions to diving into a ***crucial part*** of dataset distillation, establishing a ***standardized evaluation benchmark*** (Reviewer twQ2), offering a ***new perspective*** (Reviewer twQ2), and providing ***valuable insights into the validation*** for dataset distillation (Reviewer izfT) are acknowledged.

### Summary of Contribution and Novelty

Our work stands out through the following key contributions and innovations:

1. **Novel Perspective and Insightful Findings:**
    - To our knowledge, this paper is the **first t**o explore how to choose effective loss functions for label utilization in DD frameworks, which is recognized by Reviewers vaeM and FuA8.
    - Our study reveals the intriguing fact that models trained on synthetic datasets are ***highly sensitive*** to the choice of the loss function, which has been acknowledged as crucial and interesting by Reviewer vaeM and FuA8.
2. **Identification of a Key Yet Overlooked Challenge:**
    - We reveal that traditional loss functions suffer from significant robustness deficiencies when applied across different optimizers, a critical but overlooked challenge highlighted by Reviewer vaeM.
    - Our proposed method notably improves cross-optimizer generalization, which has been emphasized and recognized by Reviewer twQ2 and vaeM.
3. **Method with a Solid Theoretical Foundation and No Additional Costs:**
    - We propose GIFT, a simple and universal label utilization algorithm including label refinement and a cosine similarity-based loss function.
    - We provide a ***solid theoretical analysis*** supporting the proposed use of cosine similarity, as acknowledged by Reviewer FuA8.
    - GIFT builds upon existing dataset distillation methods without requiring additional information, thus raising **no extra cost**, as highlighted by Reviewer twQ2.
4. **Well-designed Experimental Validation:**
    - Our experiments encompass datasets of ***varying scales and resolutions***, from small-scale, low-resolution datasets (e.g., CIFAR-10/100) to large-scale, high-resolution datasets (e.g., ImageNet-1K). The effectiveness and practical benefits of our method are recognized by all reviewers.
    - Beyond state-of-the-art experiments, our ***strong ablation studies and applications*** in continual learning have shown promising results, as highlighted by Reviewer **FuA8.**
    - GIFT significantly enhances dataset distillation methods in both ***cross-architecture and cross-optimizer generalization.***
        - We ***empirically and theoretically*** analyze why optimizers exert a significant influence on the performance of various distilled datasets, as elaborated in Appendix E on pages 21-25 in the revised manuscript.
5. **Potential Impact:**
    - This paper shifts focus from the distillation process itself to the model training phase using synthetic data, presenting ***a novel perspective*** for dataset distillation (Reviewer twQ2).
    - The approach in this paper could potentially help ***establish a standardized evaluation benchmark*** for dataset distillation products, as recognized by Reviewer twQ2.
    - It provides ***valuable*** insights into the validation of dataset distillation, as noted by Reviewer izfT.

---

### Meta-Review · Area_Chair_3Gjj · 2024-12-19

**Metareview:**

This work comprehensive evaluates several loss functions for dataset distillation and reveals the high sensitivity of model trained on the
synthetic dataset to the choice of loss function. leveraging this finding, the authors introduce a new method called GIFT, which is a simple yet effective method for dataset differentiation with soft label refinement and a cosine similarity-based loss function. Overall, I believe the proposed GIFT method is novel, the experimental results are solid and demonstrate that GIFT significantly enhances performance without incurring additional computational costs. Therefore, I would like to suggest accept to this manuscript.

**Additional Comments On Reviewer Discussion:**

There are some minor concerns primarily regarding the performance of the proposed loss function in alternative scenarios, such as its application to hard/smoke labels (Reviewer izfT) or its integration with other related methods (Reviewer twQ2). Overall, the reviewers feel that the authors' rebuttal have well addressed these concerns and are positive towards this work.

---

### Decision · Program_Chairs · 2025-01-22

Accept (Poster)